



# Turbulent transport extraction in time and frequency and the estimation of eddy fluxes at high resolution

Gabriel Destouet[1], Nikola Besic[2], Emilie Joetzjer[1], and Matthias Cuntz[1]

[1]Université de Lorraine, AgroParisTech, INRAE, UMR SILVA, Nancy, France
[2]Laboratoire d'Inventaire Forestier, IGN, ENSG, Nancy, France

**Correspondence:** Gabriel Destouet (gabriel.destouet@inrae.fr)

**Abstract.** We propose a framework for the estimation of eddy fluxes based on cross-scalogram smoothing. This is motivated by two main problems encountered with the standard eddy-covariance method: (1) limited temporal resolution leading to estimated fluxes unable to characterise fast dynamics ($< 30\,\mathrm{min}$) and with potentially large periods of data discarded after the application of quality tests; (2) limited frequency resolution leading to poor localisation of the turbulent scales and thus to potential biases

in the estimations. We show that cross-scalogram smoothing can be viewed as an extension of the standard eddy-covariance approach where measurement signals are analysed in multiple frequency bands leading to a high resolution analysis of fluxes in time and frequency. A metric based on the vertical component of the Reynold's stress tensor is proposed to localise the turbulent scales in time and frequency. It conditions the estimation of any scalar flux decomposed in time and frequency. The proposed metric is similar to the $u^*$ and $\sigma_w$ tests but it is adapted to the time-frequency setting. We also address practical

issues encountered with cross-scalogram smoothing such as the choice of the wavelet family and the conservative property of the decomposition. We show application of the framework at the beech forest site FR-Hes and demonstrate its relation with standard eddy covariance calculations. The proposed method produces high temporal resolution (1 min) estimates of $CO_2$, latent and heat fluxes that align well with estimates from filtered 30 min time-step standard eddy-covariance method. The improved localisation of turbulent scales results in higher estimates of carbon uptake during summer ($+2\pm1\ \mu\mathrm{mol\ m^{-2}\ s^{-1}}$)

and a more accurate assessment of nighttime respiration compared to standard eddy-covariance estimates. The methodology is implemented in the Julia package TurbulenceFlux.jl and is readily available for use.

## 1  Introduction

The establishment of extensive networks of flux towers across the globe over the past two decades has proven to be a valuable asset for the scientific community and policy makers alike. It enabled monitoring of a diverse range of ecosystems and a

more detailed characterisation of their functioning (Baldocchi, 2019). This is particularly important in light of the uncertain effects of climate change on these ecosystems. In order to make the best use of the instruments and the data, standards have been established with regard to the instrumentation setup and the data processing methods. In particular, the eddy-covariance method has evolved to become a standard approach (see e.g. Burba, 2022). It is now widely employed to estimate fluxes from raw measurements and is available through different software packages (Fratini and Mauder, 2014).



Standard eddy covariance processing is constrained by its fixed time step and averaging time, typically set to 30 or 60 minutes. This limited resolution translates into large periods of estimates being discarded after quality assessment (Aubinet et al., 2012, Chap. 4) with 20% to 60% of data products likely to be rejected (Falge et al., 2001; Papale et al., 2006). Increasing the resolution is likely to reduce periods without estimates and produce fluxes with faster dynamics, thereby opening up new research opportunities, such as studying the fast responses of plants to environmental cues (Durand et al., 2021).

The standard eddy covariance method is based on Reynold's decomposition and the covariance operator. The Reynold's decomposition implies that signals are considered ergodic so that the rules of averaging apply (see Stull, 1988, Sec. 2.4.2). An important consequence of this is that a single parameter, the averaging time length, determines two key elements: (1) how signals are decomposed over time into mean and variable parts, where the latter should only capture local turbulent processes; (2) the duration and time step of the covariance operator for estimating the fluxes. However, it is clear from the experimental

studies that turbulence above canopy is an intermittent process that cannot be considered ergodic (Lee et al., 2004, Chap. 8). A direct consequence is that Reynold's decomposition and the covariance operator are regarded as filtering operations. Reynold's decomposition separates signals into low and high frequency components, corresponding to the mean and variable parts, with the averaging time determining where this separation occurs in frequency. The ideal location is within the spectral gap between mesoscale and turbulent processes (Van der Hoven, 1957; Von Randow, 2002), such that the high frequency component (the

variable part) only captures the local turbulent transport. The covariance operator estimates a flux by correlating the variable parts over a short duration which is set to the same averaging time used during the Reynold's decomposition. Two points can be remarked. Firstly, the Reynold's decomposition does not adapt to changing conditions, i.e. a changing spectral gap, which means that the variable part of signals may not contain at any given time only the information relative to a local turbulent transport, resulting in potential biases. Secondly, the averaging time of the covariance operator is dependent on the spectral gap,

yet it should only depend on the duration of the coherent structures involved in the turbulent transport. This makes it impossible to modify the operator to reach higher temporal resolutions. To overcome these limitations, an alternative decomposition that tracks the evolution of turbulence and a different estimation operator that is independent of the decomposition can be used with the goal of estimating fluxes at high temporal resolution.

    The identification and extraction of localised patterns, such as microfronts, which are local and coherent structures hypoth-

esised to contribute significantly to the flux, has been proposed to account for the intermittent nature of turbulence (Schols, 1984; Bergström and Högström, 1989). This has led to the development of numerous conditional sampling methods (see e.g. Subramanian et al., 1982) for extracting patterns from turbulence time-series. In that context, wavelet transforms appeared as an alternative to Reynold's decomposition and proved to be efficient in localising patterns (Mahrt and Frank, 1988; Mahrt, 1991). Collineau and Brunet (1993) first used conditional sampling with wavelet transform for studying the turbulent exchange

of heat and momentum between the canopy of a pine forest and the atmosphere. The identification of structures with wavelet transforms has been exploited in many other cases: energetic eddies (Howell and Mahrt, 1994) whose approach based on a energetic criteria is reminiscent of the one used by (Katul and Vidakovic, 1996) for the identification of detached/attached eddies (see Townsend, 1980); $H_2O/CO_2$ concentrated eddies for flux partitioning (Scanlon and Albertson, 2001); upwards and



downwards eddies at different heights along a shore (Attié and Durand, 2003); and large scale structures over heterogeneous
terrain (Mauder et al., 2007).

It has also been proposed to directly estimate fluxes from cross-scalograms, i.e. the product of the wavelet transforms of
two signals, instead of focusing on particular patterns. To our knowledge Attié and Durand (2003); Strunin and Hiyama (2004)
first exploited continuous cross-scalograms to form instantaneous fluxes in position-wavelength representation with airborne
measurements, which is equivalent to a time-frequency representation for static measurements. Mauder et al. (2007) used local
smoothing over cross-scalograms (see Torrence and Compo, 1998) to form averaged fluxes at different wavelengths, with their
sum over wavelengths leading to a flux estimates over a region. The local smoothing serves as a noise removal and estimates
a local correlation (i.e. a flux) in time and frequency coordinates. This method differs from the standard eddy-covariance
approach as the decomposition in different frequency bands is done separately from the actual estimation of the flux by local
smoothing. It can be seen as a generalization of the standard approach where more frequency bands are added to the Reynold's
decomposition, and where the covariance operator is replaced with a cross-correlation parameterised with a averaging length
parameter independent of the decomposition. That way, the flux is estimated locally in time and frequency coordinates but
without limitations in the time resolution and with sufficient frequency bands to better localise turbulent transport.

The smoothing of cross-scalograms thus seems a promising approach to overcome the limitations of the standard eddy-
covariance approach. However, its use may have been hindered by the presence of various obstacles. The wide variety of
available wavelet types makes selection a challenging process. The overall estimation may depend on a particular type of
wavelet (Schaller et al., 2017), which raises the question of whether another family could have been more optimal. Furthermore,
different types of wavelet transformations are possible, such as the orthogonal wavelet transform or the transformation of
signals with redundant frames of wavelet. The decomposition should, however, conserve energy and the global flux. There
is also currently no test, that is adapted to time and frequency coordinates, for determining that the turbulence is sufficiently
developed and that the estimated flux is linked to a local turbulent transport.

In the presented work, we address the previously outlined issues. First, we establish a general framework for decomposing
and estimating fluxes in time and frequency coordinates and show how the standard eddy-covariance approach or the local
smoothing of cross-scalograms can be viewed as particular cases of it. We specify the conditions for ensuring that the de-
composition and estimation process conserves the global flux. Next, we present a particular case of the established framework
by employing a redundant frame of Generalised Morse Wavelets, a parameterised superfamily of wavelets that facilitate the
exploration of various wavelet shapes (Lilly and Olhede, 2012). Finally, we develop a statistical test based on the Reynold's
stress tensor to assess the development of turbulence in time and frequency coordinates. We present results of the proposed
methodology using data acquired at the FR-Hes flux tower (Granier et al., 2008).

## 2 Methodology

We denote the vertical wind speed as $w$, the horizontal wind speeds as $u$ and $v$, the temperature as $\theta$ and a scalar of interest
such as $CO_2$ as $s$. We assume that signals are observed over a period $T$. The Fourier transform of the signal $x$ is denoted as $\hat{x}$.



## 2.1 The standard eddy-covariance approach and its limits

We recall here the standard eddy-covariance method, before showing how it can be extended in the next section. We consider the simple case of the conservation of a passive scalar $s$ in an incompressible flow with horizontal homogeneity, with the
equation of conservation being (Sec. 3.2.6 Stull, 1988):

$$\frac{\partial s}{\partial t} + \frac{\partial ws}{\partial z} = S_s, \tag{1}$$

where we neglected, for simplicity, the fluxes by diffusivity, and where $S_s$ represents the source/sink term. The standard approach uses Reynold's decomposition to split the advective term, the second term in Eq. 1, into two terms: the eddy fluxes and fluxes due to larger scale structures. It uses the averaging operator defined by:

$$\overline{x}(t) = \frac{1}{T_c} \int\limits_{t}^{t+T_c} x(\tau)d\tau, \tag{2}$$

where $T_c$ is the averaging time, and $N$ is the number of averaging periods over the observation period $T$, such that $T = NT_c$. Over a discrete time grid $t = kT_c$, so that $\overline{w}$ and $\overline{s}$ are constant over adjacent periods $[kT_c, (k+1)T_c]$, the averaged advective term is decomposed using the signal decomposition $w = w' + \overline{w}$ which results in:

$$\overline{ws} = \overline{w's'} + \overline{w}\,\overline{s}, \tag{3}$$

with the covariance operator appearing in the first term $\overline{w's'}$ because:

$$\overline{w's'} = \overline{(w - \overline{w})(s - \overline{s})}. \tag{4}$$

The averaging and decomposition of the advective term in Eq. 1 with the Reynold's decomposition at times $t = kT_c$ results in:

$$\frac{\partial \overline{s}}{\partial t} + \frac{\partial \overline{w's'}}{\partial z} + \frac{\partial \overline{w}\,\overline{s}}{\partial z} = \overline{S_s} \tag{5}$$

where $\frac{\partial \overline{w's'}}{\partial z}$ represent the eddy fluxes and $\frac{\partial \overline{w}\,\overline{s}}{\partial z}$ the fluxes due to larger scale structures.

To relate the eddy fluxes to the ecosystem fluxes $\overline{S_s}$, the storage term (first term in Eq. 5) must be taken into account along with the influence of large scale structures. The later can be neglected if no subsidence is assumed, i.e. that $\overline{w} = 0$.

The Reynold's decomposition acts as a filtering operation (Kaimal et al., 1989; Lee et al., 2004, Chap. 2), where averaged quantities $(\overline{w}, \overline{s})$ results from the application of a low-pass filter and thus contains information about large scale structures,
while the variable part $w'$ are the remaining high frequency components of $w$ likely characterizing small turbulent structures. This separation, i.e. the chosen averaging time $T_c$, should be in accordance with the spectral gap separating the turbulent scales from the larger scales. If the separation occurs outside the spectral gap then the fluxes are biased. For example if the averaging time $T_c$ is such that it falls inside the band of frequencies occupied by the turbulent scales, then $\overline{s}$ contains information about the local turbulent transport, which is lost considering only the correlation between the high frequency components $w's'$. The





low frequency components are influenced by external forcings which influences the position and width of the spectral gap throughout the day (Von Randow, 2002; Lee et al., 2004). Thus at any given time, it is unlikely that the high frequency part contains all of and only the information relative to turbulent transport. The averaging time used for the Reynold's decomposition should hence adapt dynamically to measurement conditions.

The averaging time of the standard eddy covariance approach places constraints on both, the manner in which signals are

decomposed and the estimation period used to calculate eddy fluxes via the covariance operator. The decomposition and the estimation could be parameterised independently, with the former influenced by the location of the spectral gap and the latter dependent on the time support (physical size) of the coherent structures.

## 2.2 An extension of the standard approach with a higher resolution in time and frequency

To overcome the aforementioned problems of the standard eddy-covariance, the frequency band occupied by the turbulent

scales needs to be estimated at any given time. The proposed approach is to decompose into several frequency bands and decide at any given time which portion of these frequency bands will be used to estimate the eddy fluxes.

Here, we split the advective term into more frequency bands instead of the two frequency bands originally present in the Reynold's decomposition. With $L$ such frequency bands spanning all frequencies, we denote as $w_l$ and $s_l$ the filtered versions of $w$ and $s$, where $w_l$ correspond to the analysis in the $l^{\text{th}}$ frequency band. An averaging operator $[\cdot]_\phi$ is introduced, where

$[x]_\phi = x * \phi$ is the convolution between the signal $x$ and the averaging function $\phi$. Similarly to Eq. 3, the advective term averaged here with $\phi$ is expanded using:

$$[ws]_\phi \simeq \sum_{l=1}^{L} [w_l s_l]_\phi , \tag{6}$$

We present in Appendix A5 an analysis of the viability of this approximation in function of some parameters of the decomposition that will be detailed in Sec. 2.4. Then the eddy fluxes are localised through time and frequency using this decomposition.

A "turbulence" mask is introduced $\mathcal{X}(t,l), t \in [0,T], l \in \{0,\dots,L-1\}$, where $\mathcal{X}(t,l) = 1$ indicates that the frequency band $l$ contains turbulent eddies at time $t$ and it is $0$ otherwise. The advective term is separated into turbulent eddy fluxes and other fluxes, which includes fluxes due to large scale processes and noise:

$$[ws]_\phi(t) \simeq \underbrace{\sum_{l=1}^{L} [w_l s_l]_\phi}_{} = \underbrace{\sum_{l=1}^{L} \mathcal{X}(t,l) [w_l s_l]_\phi(t)}_{\text{Eddy fluxes}} + \underbrace{\sum_{l=1}^{L} (1 - \mathcal{X}(t,l)) [w_l s_l]_\phi(t)}_{\text{Large scale fluxes + noise}} . \tag{7}$$

The conservation of mass equation writes then:

$$\frac{\partial [s]_\phi}{\partial t} + \sum_{l=1}^{L} \mathcal{X}(t,l) \frac{\partial [w_l s_l]_\phi}{\partial z} + \sum_{l=1}^{L} (1 - \mathcal{X}(t,l)) \frac{\partial [w_l s_l]_\phi}{\partial z} = [S_s]_\phi , \tag{8}$$

To relate the eddy fluxes to the ecosystem fluxes the storage term (first term) and the influence of large scale fluxes (third term) need to be taken into account.





The proposed decomposition of the advective term (Eq. 6) is similar to the decomposition made in Reynold's decomposition (Eq. 3), where the filters are the averaging operator of Eq. 2 and its high-pass filter counterpart.

The local smoothing of cross-scalograms (Mauder et al., 2007) is also a particular decomposition of the advective term. The filtered versions $w_l, s_l$ are wavelet decomposition of signals at particular scales $l$ (see Torrence and Compo, 1998). It leads to the formation of cross-scalograms with the product $w_l s_l$ and to local estimation of the flux in time and frequency coordinates through averaging with $[w_l s_l]_\phi$.

In the next section 2.3, we elaborate a general framework that presents sufficient conditions on the filters and the averaging

operator so that the decomposition into different frequency bands conserve the global flux, i.e. to verify that,

$$\int_0^T \sum_{l=0}^L [w_l s_l]_\phi (\tau) d\tau = \int_0^T w(\tau) s(\tau) d\tau. \tag{9}$$

Later, in section 2.4, we show how to implement that framework by relying on Generalised Morse Wavelets. Finally, we introduce in section 2.5 a metric based on the vertical amplitude of the Reynold's stress tensor for identifying the vertical turbulent transport and estimating the turbulence mask $\mathcal{X}$.

**2.3    A general framework for decomposing fluxes in time-frequency space**

The proposed framework chooses a set of filters $\{\psi_l\}_l$ indexed with $l \in \{0, \ldots, L-1\}$ as well as an averaging function $\phi$. These filters could be, but are not limited to, the well-known wavelets (Mallat, 2009). We start by filtering the signals with the set of filters $\{\psi_l\}_l$ leading to a decomposition in frequency bands. Each filter $\psi_l$ occupies a particular frequency band indexed with parameter $l$. The filtered versions of $w$ and $s$ are computed using:

$$w_l(t) = \int_0^T w(\tau) \psi_l(t - \tau) d\tau = (w * \psi_l)(t), \tag{10}$$

with $x * y$ denoting a convolution between the two signals $x$ and $y$.

For each frequency band $l$ and at each time $t$, a local flux in time and frequency $\mathrm{F}_s(t, l)$ is estimated using the averaging function $\phi$

$$\mathrm{F}_s(t, l) = \int_0^T w_l(\tau) s_l(\tau) \phi(t - \tau) d\tau = (w_l s_l * \phi)(t) = [w_l s_l]_\phi(t), \tag{11}$$

which is the convolution of the product $w_l s_l$ with the averaging function $\phi$ at time $t$.

We impose the following conditions on the filters and the averaging function:

**Condition 2.1.** The decomposition with filters $\{\psi_l\}$ is *self-dual* (see Mallat, 2009, Sec. 5.1.5), i.e. the energy spectral density of all filters sum to one:

$$\sum_{l=1}^L \left| \widehat{\psi_l}(\nu) \right|^2 = 1 \quad \forall \nu, \tag{12}$$





with $\widehat{\psi}_l(\nu)$ being the Fourier transform of $\psi_l$ at frequency $\nu$.

**Condition 2.2.** The averaging function $\phi$ is positive and integrates to a constant unit signal over the observation period $T$:

$$\int\limits_0^T \phi(\tau - t)d\tau = 1 \quad \forall t \in [0, T]. \tag{13}$$

A Gaussian window can, for example, be chosen as the averaging function where the variance controls the level of smoothing of the estimated fluxes.

Conditions 2.1 and 2.2 are sufficient conditions (see Appendix A1) so that the global flux:

$$F_s^T = \frac{1}{T}\int\limits_0^T w(\tau)s(\tau)d\tau, \tag{14}$$

can be recovered by summing over all filters and integrating through time $F_s(t, l)$:

$$\frac{1}{T}\int\limits_0^T \sum_{l=1}^L F_s(\tau, l)d\tau = F_s^T. \tag{15}$$

### 2.4 Time and frequency decomposition of fluxes with Generalised Morse Wavelets

Choosing a particular set of filters depends on the application and generally requires precise insights on the property of the signals under study. The turbulent process is scale invariant, and with Taylor's hypothesis of frozen turbulence it follows that fast varying oscillations are associated to eddies of small size and inversely that a slow varying signal is related to eddies of larger size (Stull, 1988; Powell and Elderkin, 1974). Wavelets share the same property, i.e. if $\psi(t)$ is a wavelet and $\psi_a(t) = \psi(t/a)/\sqrt{a}$ is a scaled version then the frequency peak of $\psi(t)$ is scaled by a factor $a$. In other words, a wavelet at small scale

captures fast varying oscillations over short periods of time and a wavelet at high scale captures fast variations. Thus the scale of a wavelet, which is proportional to the length of its time support, can be related to the physical size of eddies.

Instead of making an arbitrary choice of a particular family of wavelets such as Mexican hat or Morlet wavelets, we base our approach on Generalised Morse Wavelets (Lilly and Olhede, 2009, 2012), which is a parameterised superfamily that encompasses a wide variety of wavelets. It is a two parameter family of analytic wavelets defined in frequency by:

$\widehat{\psi}_{\beta,\gamma}(\nu) = C\nu^\beta e^{-\nu^\gamma}, \; \nu, \beta, \gamma > 0$ $\qquad$ (16)

with $C$ being a normalisation constant. $\beta$ and $\gamma$ are two shape parameters that control notably the frequency peak, the kurtosis and skewness of the wavelet spectrum (for more details see Lilly and Olhede (2009)).

The practical advantage of using Generalised Morse Wavelets is that it avoids having to choose between many different wavelet families, each with its own implementation details. Here, the shape parameters $\beta$ and $\gamma$ can be changed to adapt the

decomposition. Perrier et al. (1995) showed that $\beta$ should not be smaller than $(\alpha-1)/2$ where $\alpha$ is the exponent of the energy spectral density of the analysed signal. This gives the lower limit of 1/3 for $\beta$ if we assume the Kolmogorov-Obukhov spectrum





where $\alpha = 5/3$. We chose the parameters $\beta = 2$ and $\gamma = 3$ as they produce wavelets with a good energy concentration in time and frequency space (Lilly and Olhede, 2012) and consequently localises well different turbulent events.

Eq. 16 is used to define a mother wavelet, using the chosen shape parameters, which is upscaled iteratively to form a set of filters. Starting at the lowest scale $a_0$ of the mother wavelet, $J \cdot Q$ upscaled versions with scales $a_i = a_0 2^{i/Q}, 0 \leq i \leq JQ - 1$ are iteratively constructed with $J$ the number of octaves and $Q$ the number of inter-octaves. This leads to a set of filters whose frequency peaks are logarithmically spaced by a factor $2^{-i/Q}$ from the highest to the lowest frequency. $Q$ controls how resolved is the analysis between two octaves while $J$ controls how far goes the analysis towards the lowest frequency (which is ultimately limited by the length of the observation period). The wavelet frequency peaks are at $\nu_i = \frac{1}{a_i} (\beta/\gamma)^{1/\gamma}$. Each wavelet is normalised in frequency by the value at its frequency peak. In practice, wavelets at the highest scales may be discarded if their frequency spectrum is not well enough sampled, leading to a lower number $K < JQ$ of wavelet filters. Since Generalised Morse Wavelets are first instantiated in the frequency domain, poor sampled wavelets appears at the lower end of the spectrum thus a limiting frequency can be chosen, e.g. $\nu_{\min} = 2F_s/N$, where $F_s$ is the sampling frequency and $N$ the sample size, so that wavelets with frequency peaks below that limiting frequency are discarded. Finally a low pass filter, noted $h$ in the following, is added to our current set of wavelet filters so that the lowest frequency region not yet spanned by the wavelets is captured. We use a simple Gaussian filter for the low pass with a -3 dB cutting frequency set at the lowest frequency peak of the set of wavelet filters. This leads to a set of $L = K + 1$ filters composed of $K$ wavelets $\{\psi_{a_i} \mid a_i = a_0 2^{i/Q}, 0 \leq i \leq K - 1\}$ and a low pass filter that we note $h$. To satisfy condition 2.1, each wavelet and the low pass filter has its frequency spectrum divided by:

$$G(\nu) = \sqrt{\left|\widehat{h}(\nu)\right|^2 + \sum_{0 \leq i \leq K-1} \left|\widehat{\psi}_{a_i}(\nu)\right|^2}. \tag{17}$$

After normalisation, we obtain a set of filters that respects condition 2.1 and have the same characteristics as wavelets. The procedure keeps an important property for the analysis of turbulence: the effective scale of the filters (time support length) still varies in inverse proportion with their frequency peaks. Thus, the theoretical frequency peaks of the initial set of wavelets can still be used as proxies for the scales of the filters. We give in Appendix A2 more details on the impact of this normalisation step.

Note that the normalisation of Eq. 17, which ensures global flux conservation, is motivated by the wavelet frame theory (see Mallat, 2009, Sec. 5.1.5) that can generally be applied to any set of filters. It is different from the $C_\psi$ reconstruction constant found in Torrence and Compo (1998). The latter comes from the discretization of the admissibility condition for continuous wavelet transforms. However, for practical applications where signals are always decomposed on a discrete set of scales, wavelet frame theory applies rather than the theory of continuous wavelet transforms.

With the Reynold's frozen turbulence assumption and the property that the frequency peaks of wavelets are linked to their scale, the filtering of any signal with a wavelet $\psi$ with frequency peak $\nu$ is equivalent to an analysis at a hypothetical eddy scale $\lambda \propto |u|/\nu$ with $|u|$ the mean amplitude of the wind. Using in our case the aerodynamic height $z - d$ where $z$ is the measurement height and $d$ the displacement height, the normalised frequency is given by:

$$\eta = \frac{(z-d)}{|u|/\nu} = \frac{(z-d)\nu}{|u|}. \tag{18}$$



This normalised frequency can be interpreted as the ratio between a height above "ground" of the observations and the size of a hypothetical eddy at time $t$ and oscillating frequency $\nu$. High normalised frequencies indicate eddies of small sizes with fast oscillations, and inversely low normalise frequencies indicate large eddies with slow oscillations.

For the rest of the paper, we will drop time-vs-frequency band index coordinates such as in Eq. 11, and use time-vs-normalised frequency coordinates where $\eta$ will be the frequency peak of the wavelet covering the $l^{\text{th}}$ frequency band. For

visualisations in Sec. 3, we allowed the normalised frequency $\eta$ to be time dependent as the mean amplitude of the wind $|u|$ varies through time, thus time and frequency decompositions will be presented in Lagrangian coordinates $(t, \eta(t))$.

## 2.5  Identification of vertical turbulent transport in time and frequency

With the standard eddy-covariance method, different statistics has been proposed to assess the quality of the estimated flux (Foken, 2017, Sec. 4.3.2). Flux variance similarity (or integral turbulence characteristics), friction velocity $u^*$ and wind speed

variance $\sigma_w$ have been proposed to test turbulence development (see Foken and Wichura, 1996).

Here, an alternative approach adapted to time and frequency coordinates is proposed. The vertical amplitude of the Reynold's stress tensor is used to identify in time and frequency coordinates the vertical turbulent processes. It assesses across time and frequency the contributions by eddies in the vertical deformations and vertical momentum of an elementary volume under observation. To do so, we propose the following metric:

$$\tau_w(t, \eta) = \sqrt{F_u(t, \eta)^2 + F_v(t, \eta)^2 + F_w(t, \eta)^2}. \tag{19}$$

where $F_u$, $F_v$ and $F_w$ are the vertical kinematic fluxes computed using Eq. 11 at time $t$ and normalised frequency $\eta$.

This quantity is related to friction velocity $u^*$ and the variance of vertical wind speed $\sigma_w$. With Reynold's decomposition, it could be written as $\sqrt{\overline{u'w'}^2 + \overline{u'v'}^2 + \overline{w'w'}^2} = \sqrt{u^{*4} + \sigma_w^4}$, and Eq. 19 is an extension to a time and frequency representation with higher resolution.

Our methodology to identify time and frequency regions is based on the analysis of $\tau_w$. $\tau_w$ is large in time and frequency regions with vertical turbulent structures due to buoyancy or mechanical shear. In order to identify such regions, a threshold $\delta_\tau$ is set to find all points $(t^*, \eta^*)$ such that $\tau_w(t^*, \eta^*) > \delta_\tau$. In practice, this threshold is estimated so that most of the sensible heat $F_H(t, \eta)$ is preserved. With numerical experiments (see Sec. 3.2), we found its value to be around $10^{-3}\,\mathrm{m}^2\,\mathrm{s}^{-2}$. This first step identifies all time and frequency coordinates at which vertical processes have an impact on the volume of air measured,

that means in removes noise. However it is also possible that slow-varying trends may be present, potentially originating from inhomogeneities in the scalar field being advected or by subsidence. An additional step is necessary to remove these unwanted regions in time-frequency space. In the spectral analysis of turbulence, this step is equivalent to finding a spectral gap between the turbulent scales and the contributions of larger scales (Powell and Elderkin, 1974). The main difference here is that we estimate a *time-dependent spectral gap* that is adapted to non-stationary settings.

We analyse the Laplacian (second derivatives) of $\log \tau_w$ to find the separating spectral gap. High values of the Laplacian suggest the presence of minima in regions that are primarily affected by noise and spurious correlations. We expect that such regions exist in-between the turbulent scales and the larger scales. We identify all time and frequency points





$\mathcal{U} = \{(t, \eta) \mid \Delta \log \tau_w(t, \eta) > \delta_{\Delta \tau_w}\}$ with a Laplacian larger than a given threshold $\delta_{\Delta \tau_w}$. We fit a curve to the selected points $\mathcal{U}$ with a locally weighted regression method (Cleveland and Devlin, 1988), which is the time-varying spectral gap $\eta_*(t)$ sepa-

rating the time and frequency regions with local turbulent transport from the influence of larger scale structures. We found that the method is not very sensitive to the value of the threshold $\delta_{\Delta \tau_w}$ and set it to 1 here. It has to be sufficiently small to select a sufficient number of points such that the locally weighted regression method passes preferably in regions with a high number of detected minima.

The above steps lead to the creation of the turbulence mask introduced in Eq. 8, identifying time-frequency regions with

sufficiently developed turbulence:

$$\mathcal{X}(t, \eta) = \begin{cases} 1, & \text{if } \eta > \eta_*(t) \text{ and } \tau_w(t, \eta) > \delta_\tau \\ 0, & \text{otherwise.} \end{cases} \tag{20}$$

This "turbulence" mask covers the time and frequency regions with normalised frequencies above the estimated time-dependent spectral gap $\eta_*$ and with sufficiently strong vertical kinematic fluxes. Note that the turbulence mask is indexed with time and normalised frequency $(t, \eta)$ here while it is indexed with time and the index of the band of frequency $(t, l)$ in Eq. 8. We relate

$\eta$ to the frequency peak of the wavelet covering the $l^{\text{th}}$ frequency band.

## 2.6    Summary of proposed methodology

We depict in Fig. 1 a summary of the proposed methodology. First, a frame of wavelets with conservative property is used to decompose wind speeds and other scalars in time and frequency. Cross-scalograms are formed as the product of the signal decompositions, and time averaged to form fluxes resolved in time and frequency. Vertical kinematic fluxes in time and fre-

quency coordinates are used to localise the time-frequency regions with local turbulent transport. This lead to the creation of a "turbulence" mask that is applied then over time-frequency decomposed scalar fluxes to integrate time resolved scalar fluxes.

The approach requires several parameters: 1. $\beta$ and $\gamma$ determine the shape of the Generalised Morse Wavelet. 2. $J$ and $Q$ control the overall resolution in frequency of the decomposition. 3. $\sigma$ is the width of the averaging function. 4. The two thresholds $\delta_{\tau_w}$ and $\delta_{\Delta \tau_w}$ condition the identification of the local turbulent process in time-frequency coordinates.

The wavelet parameters need to be shared across all time and frequency decompositions to keep the same coordinate system. The averaging parameter $\sigma$ may be different in the computation of $\tau_w$ for the identification of the turbulence mask, and in the final estimation of the scalar fluxes. For example, the turbulence mask may be inferred from slow varying vertical kinematic fluxes, e.g. using $\sigma = 30$ min. The fluxes themselves can be calculated using less smoothing, e.g. using $\sigma = 10$ min.

## 3    Results

The methodology is applied on data acquired at the FR-Hes flux tower (Granier et al., 2008), a class 1 ecosystem station of the Integrated Carbon Observation System (ICOS). The analysis was performed over the whole year 2022. Here we present selected days and statistics over 8-hour periods at day and night during summer (10-19 June 2022) and winter (3-11 March





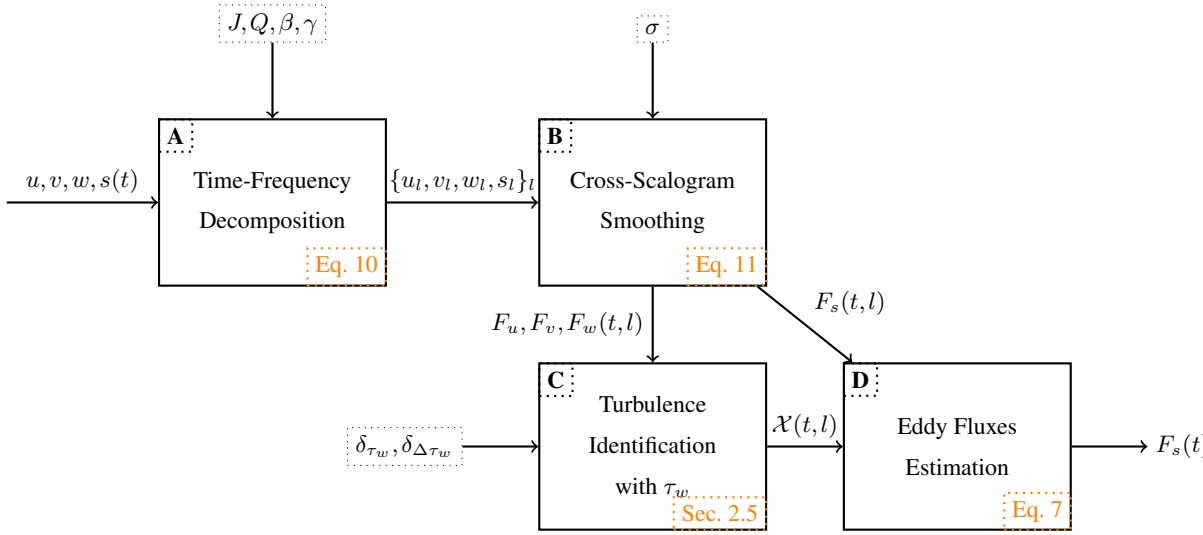

**Figure 1.** Summary diagram of the proposed methodology for estimating fluxes from turbulent transport. **A**: Signals are decomposed in time-frequency space using a set of filters composed of wavelets and a low-pass filter. They are initialised and normalised according to Eq. 17 to have conservative property. **B**: Cross-scalograms are formed and averaged through time using an averaging function of size $\sigma$ respecting condition 2.2. **C**: Given vertical kinematic fluxes the metric $\tau_w$ of Eq. 19 is computed and a turbulence mask $\mathcal{X}$ identifying the vertical turbulent transport is inferred. **D**: The flux of scalar $s$ is estimated by integrating through frequencies the vertical flux $F_s(t,l)$ in time and frequency coordinates according to the turbulence mask $\mathcal{X}$.

2022). The tower is situated in a beech forest with a roughly flat terrain ($< 3\%$ slope). The ICOS standard instrumentation (see Rebmann et al., 2018) is installed at FR-Hes with a LI-7200rs gas analyser (LI-COR Biosciences, Lincoln, USA) and a HS-50

anenometer (Gill Instruments Ltd, Lymington, UK). All signals are acquired at $20\,\mathrm{Hz}$ at a height of $30.8\,\mathrm{m}$ above ground with a canopy height of about $21.5\,\mathrm{m}$ in 2022. The displacement height was estimated to be $14.7\,\mathrm{m}$ using 2/3 of canopy height Raupach (1994). These heights are used to compute normalised frequencies (Eq. 18). All fluxes were calculated in batches of $24\,\mathrm{h}$ raw data, either centered around noon or around midnight. The filtering with wavelets and averaging creates errors at the borders of the observation period (see Torrence and Compo, 1998). The standard deviations in time of the wavelets and the

averaging filter are summed for each frequency band to estimate the erroneous time frame at the edges of the $20\,\mathrm{h}$ periods. We take the maximum across frequency bands of the standard deviations to get the size of time window to remove from the results. The time period is estimated to be $2\,\mathrm{h}$ at each side, which results in flux estimates of $20\,\mathrm{h}$ for each $24\,\mathrm{h}$ observational period.

All time and frequency decompositions are computed using the wavelets of Sec. 2.4. The shape parameters for the Generalised Morse Wavelets are taken as $\beta = 2$ and $\gamma = 3$, as explained before. The wavelets are positioned according to their

frequency peaks. Since data are acquired at 20 Hz, the frequency peak of the first wavelet is initiated at $10\,\mathrm{Hz}$, subsequent wavelets are scaled up by a factor $2^{1/Q}$ with $Q = 4$ until the minimum frequency peak of $F_s 2/N$ is reached, where $N$ is the number of samples and $F_s$ is the sampling frequency of $20\,\mathrm{Hz}$. With $24\,\mathrm{h}$ long observation periods and a sampling frequency of $20\,\mathrm{Hz}$, the limiting frequency corresponds to a time period of $12\,\mathrm{h}$. The low-pass filter then spans the remaining frequency





band below that limiting frequency thus capturing processes with oscillations periods period larger than $12\,\mathrm{h}$. Since we are mostly interested in the study of eddy fluxes here, this frequency band is always discarded. Different sizes of the averaging function are used to compute the metric $\tau_w$ and to estimate the fluxes in time (see Eq. 11 and Sec. 2.5). We choose a Gaussian window with deviation $\sigma = 30\,\mathrm{min}$ for $\tau_w$ and $\sigma = 10\,\mathrm{min}$ for the estimation of the scalar fluxes. After averaging, i.e. after step **C** in Fig. 1, fluxes in time and frequency coordinates have a temporal resolution of $20\,\mathrm{Hz}$ and around 70 frequency bands (depending on the original size of the signals). The decompositions are sub-sampled at a $1\,\mathrm{min}$ time interval to save memory. In the following, the fluxes will be presented in time versus normalised frequency coordinates $(t, \eta)$, instead of time versus the index of a frequency band $(t, l)$ as in Eq. 8, where the normalised frequency is the frequency peak of the wavelet covering the $l^{\mathrm{th}}$ frequency band.

We compare results obtained using our method (HR-TM) to the estimations without turbulence mask (HR) corresponding to a wavelet-based estimated flux that integrates over all scales, as well as to the estimations with the standard eddy-covariance approach with $30\,\mathrm{min}$ time averaging (EC30). For information and to better understand the presented results, the flux estimation with the 30 min standard eddy-covariance is roughly equivalent to integrating all the flux from the normalised frequency $\eta = 5 \cdot 10^{-3}$, if a time length of order $(z - d)/|u| \simeq 10\,\mathrm{s}$ (see Eq. 18) is assumed.

All fluxes are presented without frequency corrections in amplitude (see Burba, 2022), with $H_2O$ and $CO_2$ concentration signals being corrected for time lags with wind anemometer data. $H_2O$ and $CO_2$ concentration signals were shifted to maximize total correlation with vertical wind velocity for each $24\,\mathrm{h}$ period by using only data in the $0.1\,\mathrm{Hz}$ to $1\,\mathrm{Hz}$ frequency band. This range was chosen to reduce the potential influence of low frequency trends in the estimation of the time lag transporting at the same time sufficient information about the turbulent transport. It corresponds roughly to the normalised frequency range $\eta = 1$ to $\eta = 10$. The estimated time lags were $260 \pm 125\,\mathrm{ms}$ for $CO_2$ and $400 \pm 130\,\mathrm{ms}$ for $H_2O$ on average. Sensible heat fluxes use the so-called sonic temperature from the anemometer. It is thus not corrected for humidity (see p.42 Dijk et al., 2004). Frequency correction and humidity correction are foreseen to be included into the method in the near future. Finally, the WPL correction (Webb et al., 1980) is not required in our case as the LI-7200rs gas analyser outputs dry mole fractions (see Burba, 2022, Sec. 4.7).

The methodology has been implemented in the Julia package TURBULENCEFLUX.JL.

### 3.1 Detailed example of the estimation of $CO_2$ flux

We present in Fig. 2 a detailed example of the estimation of $CO_2$ flux. The example is for the 15th of June 2022, which was characterised by sunny conditions without precipitation. The $\tau_w$ metric (Sec. 2.5) in Fig. 2A identifies the vertical turbulent structures in time and frequency here obtained with an averaging length of $30\,\mathrm{min}$. Time and frequency regions of high intensity $(> 10^{-2}\,\mathrm{m^2\,s^{-2}})$ are approximately located from $\eta = 10^{-2}$ to $\eta = 10$, with a noticeable difference between day and night as well as the times of sunrise and sunset. This difference can be explained by the type of turbulence at play: from $6\,\mathrm{h}$ to $18\,\mathrm{h}$ the turbulence is mostly created by buoyancy within the range $\eta = 10^{-2}$ to $\eta = 1$, while from $20\,\mathrm{h}$ to $4\,\mathrm{h}$ the turbulence originates from mechanical shear and is located in the range $\eta = 10^{-1}$ to $\eta = 10$. Large regions of high intensity from $\eta = 10^{-2}$ to $\eta = 10$ indicates some stability in the physical process at play. Below $\eta = 10^{-1}$ at night and $\eta = 10^{-2}$ during the day, $\tau_w$ has generally



**Figure 2.** Turbulent transport identification and estimation of $CO_2$ flux on the 15 of June 20222. All time and frequency analyses are visualised in normalised frequencies $\eta$ versus time over 24 h. **A**: time and frequency decomposition of the amplitude of the vertical component of the Reynold's stress tensor $\tau_w$. **B**: Laplacian of $\log \tau_w$ in time and frequency space with an estimated time-varying spectral gap (dashed line) separating the turbulent transport at small scales from structures at larger scales. **C**: $CO_2$ flux in time and frequency space with a turbulence mask derived from analyses of A and B. A time varying 30 min spectral gap associated to the frequency separation of the standard eddy-covariance EC30 is also shown (solid line). **D**: High time resolution $CO_2$ flux (HR-TM) along with standard 30 min eddy-covariance (EC30).





low amplitude ($10^{-4}\,\mathrm{m^2\,s^{-2}}$) with isolated and small time-frequency regions of medium intensity ($10^{-3}\,\mathrm{m^2\,s^{-2}}$). Below these ranges in $\eta$, large-scale structures ($\eta \simeq 10^{-3}$) intermittently apply vertical stress over short periods of time ($< 2\,\mathrm{h}$).

Similar patterns were observed on different days throughout the year at FR-Hes, which motivated the following assumptions: (1) vertical turbulent transport shows large and coherent regions of high intensity in the metric $\tau_w$ at frequencies corresponding to eddies with sizes around the distance of the sensor to the roughness elements, i.e. from the measurement height to the displacement height and hence around normalised frequencies of $\eta = 1$; (2) a region of low amplitude exists in $\tau_w$ at small normalised frequencies and hence large eddies, i.e. there is a spectral gap between the turbulent region of (1) and a region

influenced by large scale processes; (3) small and isolated time-frequency regions of medium intensity at large scales ($\eta < 10^{-2}$) are considered too unstable and the scale is too large that they could be part of local vertical turbulent transport.

   Hence the high intensity regions of (1) are identified using a threshold $\tau_w > 10^{-3}\,\mathrm{m^2\,s^{-2}}$, as shown by the dotted line in Fig. 2B. The spectral gap in time-frequency space is identified by the maxima of the Laplacian of $\log \tau_w$ (see Sec. 2.5), as can be seen by the dashed line in Fig. 2B. It rejects most of the small size and medium intensity regions situated below the

spectral gap in $\tau_w$ (Fig. 2A). Together they define the mask identifying local turbulent transport relevant for estimation of ecosystem fluxes, which is the bright region in Fig. 2C. The mask is shown on top of the $CO_2$ flux decomposed in time and frequency space, highlighting the time and frequency regions with sufficiently developed local turbulence. Note that the $CO_2$ flux smoothed cross-scalogram used an averaging kernel of 10 min compared to 30 min for the determination of the turbulence mask. One can see that the mask covers time-frequency regions of high amplitude of positive and negative $CO_2$ fluxes. The

turbulence mask highlights the $CO_2$ respiration process at night and $CO_2$ assimilation during daytime. It rejects regions of large $CO_2$ fluxes at small frequencies around $\eta = 10^{-3}$ mostly from dusk into the night. The $CO_2$ flux in time and frequency coordinates is integrated along the frequency scale leading to high-resolution $CO_2$ fluxes shown in Fig. 2D. We obtain a high resolution estimation (HR-TM) that follows well the standard eddy-covariance estimation (EC30). In practice, the standard eddy covariance with 30 min resolution EC30 is approximately equivalent to integrating all the $CO_2$ fluxes from $\eta = 10^{-2}$

to $\eta = 10^2$. This is why in this particular case one does not see the influence of the $CO_2$ fluxes at larger scales ($\eta = 10^{-3}$) at night in the EC30 estimates. It also looks like that (EC30) misses some carbon uptake in the middle of the day due to the fixed integration time of 30 min. Both methods estimate a negative $CO_2$ flux at 3 h at night. This comes from the small time and frequency region of negative $CO_2$ flux at $\eta = 10^{-2}$ to $\eta = 10^{-1}$. The effect is more pronounced in (EC30) but also our method is not inert against this intermittency. If the turbulence mask does not cover any frequency bands at a given time, i.e.

no turbulence is detected at that time, the calculated flux is undefined. This can be seen around $19\,\mathrm{h}$ in Fig. 2D, where HR-TM is undefined due to the lack of frequency bands covered by the turbulence mask in Fig. 2C.

## 3.2   Statistics of $\tau_w$ and time and frequency decomposed sensible heat flux

We show in Fig. 3 probability densities of $\tau_w$ against $\eta$ and against sensible heat flux $F_H$ during day and night periods in summer. The probability densities are estimated with a kernel density estimation method. The evolution of $\tau_w$ against $\eta$

presents similar characteristics as the spectrum of the vertical velocity and as the cospectrum of $u \cdot w$ (Kaimal and Finnigan, 1994, Sec. 2.5 and Sec. 2.9). During daytime and unstable stratification (Fig. 3A), $\tau_w$ has a concave shape from $\eta = 10^{-3}$ to

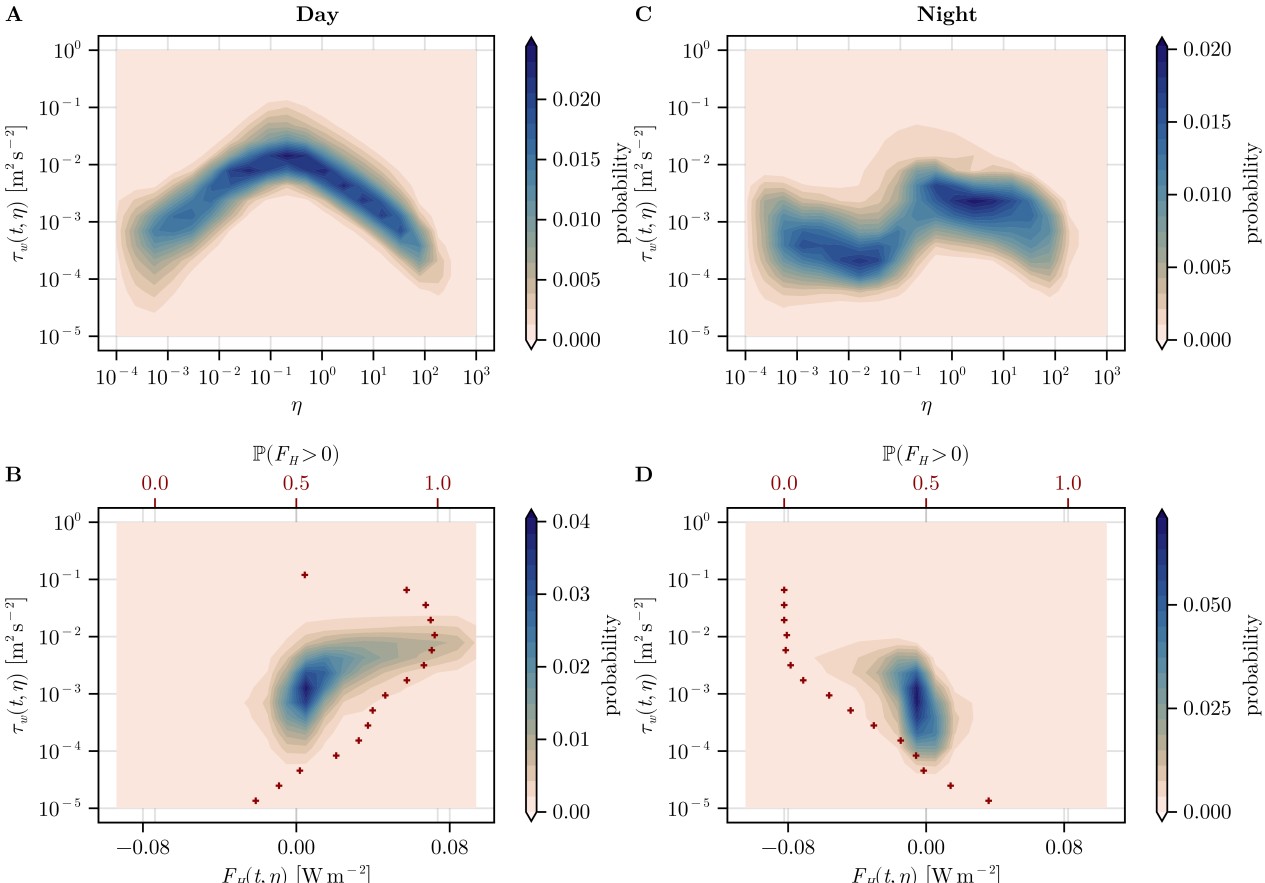

**Figure 3.** Probability densities of the amplitude of the vertical component of the Reynold's stress tensor $\tau_w(t,\eta)$ against normalised frequency $\eta$ (top row) and against the sensible heat flux $F_H(t,\eta)$ (bottom row), estimated from 10 to 19 June 2022 via kernel densities from 8 h data during daytime (left column) and during night (right column). Red crosses show the empirical probability function that sensible heat is positive ($\mathbb{P}(F_H>0)>0.5$) or negative ($\mathbb{P}(F_H>0)<0.5$) (upper x-axis) vs. $\tau_w$.



$\eta = 10^2$ with a maximum around $\eta = 10^{-1}$. During nighttime and stable stratification (Fig. 3C), the distribution of $\tau_w$ against $\eta$ has its concave shape moved to higher frequencies with a maximum around $\eta = 1$ and located from $\eta = 10^{-1}$ to $\eta = 10^2$. In comparison with daytime conditions, the nightime density of $\tau_w$ against $\eta$ presents a greater variability, it has a stronger spread

along $\tau_w$ and a less clearly defined shape. In particular, the range $\eta = 10^{-4}$ to $\eta = 10^{-1}$ is suspected to be influenced by large scale structures with a random behavior. Note that our visualisations of $\tau_w$ against $\eta$ are not normalised against a quantity such as the friction velocity nor it is weighted by the frequency as it is usually done in studies of spectra and cospectra of turbulence.

The distribution of sensible heat is mainly positive during day and the density covers mainly positive sensible heat when above $\tau_w \simeq 10^{-3}\mathrm{m}^2\,\mathrm{s}^{-2}$ (Fig. 3B). The distribution is mainly negative during night and the density covers mainly negative

sensible heat when above roughly $\tau_w \simeq 10^{-3}\mathrm{m}^2\,\mathrm{s}^{-2}$ (Fig. 3D). Choosing hence the noise threshold $\delta_\tau = 10^{-3}\mathrm{m}^2\,\mathrm{s}^{-2}$ allows clear extraction of the time-frequency regions characterizing turbulent transport of heat during day and night.

We present additional figures on the effect of applying the turbulence mask on time and frequency decomposed fluxes in Appendix A4, during daytime (Fig. A3) and nighttime (Fig. A4), with the additional densities of latent heat versus carbon fluxes. By looking at the differences in the probability densities of the data before and after applying the turbulence mask we

remark that the application of the turbulence mask (1) leads to the exclusion of large scale structures with relatively high $\tau_w$ amplitude around $\eta = 10^{-4}$, especially at night (Fig. A4G); (2) correctly preserves the turbulent exchange of heat during day and night (Fig. A4H, A3H); (3) carbon respiration, photosynthesis and evapotranspiration processes are clearly visible in the estimates (Fig. A4I, A3I). At night in particular, the application of the turbulence mask removes negative latent heat fluxes and carbon uptake (Fig.A4I), which are considered as noise and likely caused by large scales structures (see Scanlon and Sahu,

2008, Fig. 3).

## 3.3 Results in different conditions

Fig. 4 illustrates results of the methodology across four days, characterised by different conditions: two days in spring in sunny (2022-06-15, Fig. 4A) and cloudy (2022-05-01, Fig. 4C) conditions, and two days in winter in sunny (2022-02-12, Fig. 4B) and cloudy conditions (2022-01-26, Fig. 4D). For each day, we compare the estimations of latent heat with our methodology

(HR-TM) against estimations in time and frequency space without a turbulence mask (HR), i.e. including large-scale contributions down to $\eta \simeq 2{\cdot}10^{-4}$, and against the standard at 30 min eddy-covariance estimations (EC30), which corresponds roughly to integrate the flux above $\eta \simeq 5{\cdot}10^{-3}$. The estimation without a turbulence mask (HR) is equivalent to cross-scalogram smoothing without an identification of turbulence in time-frequency space. The HR flux results thus from integration over all frequencies without the first frequency band spanned by the low-pass filter (see Sec.2.4).

The EC30 and HR-TM estimations of the latent heat flux are roughly equivalent over the four selected days. However, HR can produce highly biased estimates such as on 2022-02-12 (Fig. 4B) with negative peaks in latent heat of about $-150\mathrm{W}\,\mathrm{m}^{-2}$ during the day or on 2022-05-01 (Fig. 4C) with negative fluxes around $-15\,\mathrm{W}\mathrm{m}^{-2}$. Estimations from simple cross-scalogram smoothing HR hence cannot be used without proper filtering in time-frequency space. The proposed method of using a turbulence mask on top of cross-scalogram smoothing is one way to ensure that eddy fluxes are properly estimated and that the

influence of external processes and noise are removed.





**Figure 4.** Estimation of latent heat flux on four selected days in different conditions: sunny conditions (left column), cloudy conditions (right column), during spring (top row), and in winter (bottom row). For each day: incoming short wave radiations (top), time and frequency decomposed latent heat flux (middle), flux estimates through time (bottom). 10-min high time resolution fluxes are shown using the turbulence mask (solid line, HR-TM) or integrating over all frequencies without mask (dotted line, HR). The standard eddy-covariance estimations over 30 minutes (EC30) are shown as circles. They are open symbols if they would be filtered out by a $u^*$ threshold of $0.2\,\mathrm{ms}-1$. $u^*$ (red triangles) is shown on the left axis of the bottom panels. Note that all y-axes have different ranges to clearly show the differences between the fluxes on the different days.



The $u^*$ filtering and the turbulence mask are overall in agreement, i.e. $u^*$ is high when the turbulence mask covers large time frequency regions with strong flux amplitude and $u^*$ is low when no time frequency regions are covered or when it covers regions of very low flux amplitude. We remark some exceptions such as on day 2022-06-15 (Fig. 4A) during afternoon where two half-hour estimates of EC30 would be flagged while the turbulence mask assesses that there is enough turbulence for a good flux estimation.

The dynamic of the incoming short wave radiations signal is clearly reflected in HR-TM. The effect of the passing of clouds at noon on 2022-05-01 (Fig. 4C) reduces the latent heat flux, followed by a slow increase of sunlight and latent heat from 12 h to 16 h. Due to the high resolution of the proposed approach, fast dynamic processes in the turbulent fluxes can be observed. The relation between these turbulent fluxes and the sources and sinks of the ecosystem must be analysed separately.

## 3.4 Statistics over 8h periods

We show in Fig. 5 statistics on the estimations of $CO_2$ fluxes over 8-hour periods during day and night for 9 to 10 day periods in spring (10-19 June 2022) and winter (03-11 March 2022). We show the inter-decile range and the mean of the estimated fluxes from our method (HR-TM) against the standard eddy-covariance approach (EC30).

HR-TM estimates are approximately in agreement with EC30 on average. Their spreads are of the same order of magnitude in summer but are larger for EC30 in winter during daytime (Fig. 5B). HR-TM estimates show larger carbon uptake by $+2\pm1\ \mu\mathrm{mol}\,\mathrm{m}^{-2}\,\mathrm{s}^{-1}$ during daytime in summer (Fig. 5A). This is explained by EC30 likely missing some carbon uptake below and around the scale $\eta = 10^{-2}$. During daytime in winter (Fig. 5B), both methods give weak and positive fluxes on average but contain sometimes also negative fluxes. Estimations of $CO_2$ fluxes at night are close between HR-TM and EC30 in summer and winter (Fig. 5C-D). Fluxes should be positive at night. HR-TM rarely shows negative fluxes at night, which is much more common in EC30 (Fig. 5C-D). The fixed averaging length of EC30, typically set to 30 minutes, is not well-suited for capturing the turbulent exchange driven by mechanical shear at night. This interval is typically set for turbulent exchange under unstable stratification, whereas the turbulent spectra at night shifts towards higher frequencies. As a result, nighttime EC30 estimates are corrupted in the normalized frequency range $\eta = 5 \cdot 10^{-3}$ to $\eta = 10^{-1}$, due to external processes that increase variability in the estimates, often leading to non-physical negative carbon fluxes during the night.

In Appendix A3, additional figures are presented for sensible and latent heat fluxes. For sensible heat fluxes (Fig. A2a), both the mean and the spread of EC30 and HR-TM estimates are close. For latent heat fluxes (Fig. A2b), EC30 and HR-TM estimates are approximately in agreement on average. HR-TM estimates show smaller latent heat fluxes by -37±7 $\mathrm{Wm}^{-2}$ during daytime summer (Fig. A2bA). Also, while latent heat fluxes should be positive, we remark that at night EC30 is more likely to produce negative estimates (Fig. A2bC-D).

## 3.5 Discussion

The proposed methodology allows the estimation of eddy fluxes at high time resolution taking into account local turbulence. At the FR-Hes site, the estimations are overall in accordance with estimates made by the standard eddy-covariance method in 30-min intervals. However, we suspect the standard eddy-covariance approach to be more likely biased by external low





**Figure 5.** Daytime (left column) and nighttime (right column) $CO_2$ fluxes of consecutive days with high irradiance and no precipitations. Top row: 10-19 June 2022; bottom row: 03-11 March 2022. The error bars on EC30 and the shaded areas of HR represent the 10th to 90th percentile range. Note that the y-axes have different ranges to show differences between the flux estimates on the different days.





frequency disturbances. The fixed averaging length induces a decomposition that is not adapted for nighttime conditions with
mostly mechanical shear and should be made smaller than during the day. Fluxes might also be biased during the day when
low-frequency components below $\eta = 10^{-2}$ may be relevant turbulent fluxes.

The presented framework is a generalisation of the standard eddy-covariance approach, which employs more frequency
bands to calculate fluxes and provides estimations continuously through time. The time-frequency decomposition of fluxes
allows a finer analysis of the behaviour of the turbulent transport, and the proposed metric for assessing the development of
turbulence allows a more precise localisation in time-frequency space. In particular, it enables the estimation of a time-varying
spectral gap separating the turbulent scales from larger-scale processes. As a direct consequence, the proposed approach re-
moves some biases that are apparent in the standard eddy-covariance method. This is particularly the case if the high frequency
components (the variable parts) obtained via Reynold's decomposition contain noise or large scale processes not related to the
turbulent transport of interest.

Estimations of eddy fluxes have been made above a forest ecosystem, and to relate these to ecosystem fluxes, an additional
analysis of the coupling of turbulence below and above the canopy should be made. The present methodology does not assess
this coupling and another step is required for filtering the flux only when a strong coupling is present. By integrating $\tau_w$ along
the frequencies with the turbulent mask found, it is possible to derive an alternative metric in time equivalent to $u^*$ or $\sigma_w$. It
would provide a summary metric of $\tau_w$ through time along with the fluxes, which could subsequently be used to study coupling
or the representativeness of the turbulent fluxes. This development is out of the scope of this paper but is subject of ongoing
research.

The sensitivity of our methodology for flux estimation against wavelet parameters is not presented in this work. However, we
have identified three important factors for the decomposition of fluxes into time and frequency coordinates: 1. the conservation
of the global flux, which is guaranteed by the normalization step (see Eq. 12 and 17); 2. the frequency resolution of the
decomposition ($J$ and $Q$ parameters of Sec. 2.3); and 3. time-frequency localisation of the wavelet used for the decomposition
($\beta$, $\gamma$). The normalisation (1) guarantees that the fluxes obtained after integrating the time-frequency decomposition have
meaningful physical interpretation. In particular, we do not have to estimate any wavelet-reconstruction factor empirically as
encountered with continuous wavelet transforms (e.g. Schaller et al., 2017). This means that we can safely apply the proposed
methodology for decomposing signals and estimate flux quantities. It is important to have a sufficient number of frequency
bands to separate the turbulent scales from the larger scales, thus a high frequency resolution (2) is needed and $J$ and $Q$ are
set to adequately high values. The time frequency localisation of the wavelets has also an impact on how the information
about the turbulent transport is scattered across the time-frequency plane (3). The shape of the wavelet conditions the ability
to localise events in time and frequency coordinates. Thus, we follow the recommendations of Lilly and Olhede (2012) for
setting $\gamma$ around 3 and $\beta > 1$. Values away from these parameters tend to create wavelets with poor localisation and hence
risk producing poor time and frequency decomposition. These factors have also their importance in the decomposition of
the advective term in the conservation equation Eq. 8. In Appendix A5, an analysis of the influence of the decomposition
parameters on the approximations of the advective term in Eq. 6 is presented.



The current framework enables adjusting the averaging window for estimating the flux of any scalar in HR-TM without affecting the decomposition across different scales. In this study, we presented results using an averaging window of $\sigma = 10$ min for fluxes, while the turbulence mask was determined with $\sigma = 30$ min. A more precise setting of the averaging size could be informed by analyzing the correlation functions of velocities and scalars. For instance, Lenschow et al. (1994) suggested setting an averaging size larger than the integral time scale, estimated by assuming an exponential model for the auto-correlation function of velocity. More recent studies, based on the random sweeping hypothesis of Kraichnan (1964), have shown through theoretical development (Wilczek and Narita, 2012), controlled experiments (Poulain et al., 2006; He and Tong, 2011), and simulations (Wilczek et al., 2014) that the velocity auto-correlation function exhibits more complex behaviour in both space and time. Specifically, for small time delays $\tau$, the auto-correlation function behaves as a Gaussian in the term $\tau k v_0$, where $k$ is the wavenumber and $v_0$ represents the large-scale velocity fluctuations (sweeping velocity). This suggests a potential lower bound for the averaging window, $\sigma_{\min} = (k_{\min} v_0)^{-1}$, where $k_{\min}$ is the lowest wavenumber below which no turbulence is assumed. However, determining the appropriate averaging window to connect eddy fluxes to ecosystem-scale fluxes is beyond the scope of this study and requires further investigation. This specific issue may be addressed through Large-Eddy Simulations or Direct Numerical Simulations.

The measurement acquisition systems (anemometer, gas analyser) have transfer functions that induce errors in the measurements. A theoretical inverse transfer function $T_s$ can be applied to reduce flux errors (Aubinet et al., 2012, Sec. 4.1.3). This transfer function can directly be taken into account during the normalisation step of the filters. With $T_s$ the total transfer function of the acquisition system for the scalar $s$, the new normalisation function of Eq. 17 would become:

$$G(\nu) = \sqrt{T_s(\nu)} \sqrt{\left|\widehat{h}(\nu)\right|^2 + \sum_{0 \leq i \leq L} \left|\widehat{\psi_{a_i}}(\nu)\right|^2}. \tag{21}$$

Flux corrections can also be applied afterwards by directly weighting in time-frequency coordinates the time-frequency decomposed fluxes.

## 4 Conclusions

In this paper, we presented a general framework for identifying turbulence and decomposing fluxes in time-frequency space, with its application demonstrated using the Generalised Morse Wavelets. The presented results arise from applying this approach at the FR-Hes site and are compared with the standard 30 min eddy-covariance approach. We have addressed some of the difficulties mentioned in the state-of-the-art regarding the use of wavelets, by ensuring flux conservation and offering flexibility in the parameterisation. The proposed method, while largely in agreement with the estimates obtained by the standard approach, enables high resolution estimates. It opens up new research perspectives, in particular the analysis of ecosystem response to rapid environmental changes (< 1 hour). We have made the current method available as a Julia software package (TURBULENCEFLUX.JL), with the aim of providing a useful tool for the community to develop new time-frequency approaches to flux estimation.



*Code availability.* The Julia package TURBULENCEFLUX.JL implements the proposed methodology and it is available at github.com/gabdst/
515  TurbulenceFlux.jl. Example data and notebooks are also available there.





## Appendix A

### A1 Sufficient conditions on filters and averaging function for global flux conservation

We demonstrate here that the self-dual property of the filters (Cond. 2.1) and the normalization property of the averaging function (Cond. 2.2) are sufficient conditions to preserve the global flux (Eq. 14). Assuming $L$ filters are used for the decomposition, we get:

$$\frac{1}{T}\int_0^T\sum_{l=1}^L (w_l s_l * \phi)(u)du = \frac{1}{T}\sum_{l=1}^L\int_0^T w_l(t)s_l(t)dt, \text{ with Condition 2.2} \tag{A1}$$

$$= T\sum_{l=1}^L\sum_p \widehat{w_l}(p)\widehat{s_l}(p)^*, \tag{A2}$$

with $\cdot^*$ the complex conjugate operator and using Parseval's formula (see Mallat, 2009, Thm. 2.3),

$$= T\sum_{l=1}^L\sum_p \widehat{w}(p)\widehat{s}(p)^*\left|\widehat{\psi_l}(p)\right|^2 = T\sum_p \widehat{w}(p)\widehat{s}(p)^*, \text{ with condition (Cond. 2.1)} \tag{A3}$$

$$= \frac{1}{T}\int_0^T w(t)s(t)dt = F_s^T. \tag{A4}$$

### A2 Impact of normalization on the frame of wavelets

We analyse the impact of self-dual normalisation presented in Eq. 17 on an initially constructed frame of wavelets $\{\psi_l\}_l$ from which some properties are theoretically known such as the frequency peak.

We drop temporarily the $\widehat{\cdot}$ notation indicating that we are working with Fourier transforms, and we note $\psi(\nu)$ the Fourier transform of the wavelet $\psi$ at frequency $\nu$. With $\widetilde{\psi_l}(\nu) = \psi_l(\nu)/G(\nu)$ our set of wavelet filters with self-dual normalisation $G = \sqrt{\sum_l |\psi_l|^2}$. We do not take into account the low pass filter here. The total derivative is given by:

$$d\widetilde{\psi_l} = \frac{d\psi_l}{G}\left(1 - \left|\frac{\psi_l}{G}\right|^2\right) - \sum_{j\neq l}\left|\frac{\psi_j}{G}\right|^2\frac{d\psi_j}{G} \tag{A5}$$

**Impact on frequency peaks**: In order to study the impact of the normalisation we analyse the derivative of $\widetilde{\psi_l}$ against $\beta$. Let $\psi_l = \psi_{a_l}$ be a wavelet (here its Fourier transform) with frequency peak normalization. We can show that:

$$\frac{\partial\psi_{a_l}(\nu)}{\partial\beta} = \psi_{a_l}(\nu)\log\frac{a_l\nu}{(\beta/\gamma)^{1/\gamma}} \tag{A6}$$

At the frequency peak $\nu_l = \frac{1}{a_l}(\beta/\gamma)^{1/\gamma}$, we have $\frac{\partial\psi_{a_l}(\nu_l)}{\partial\beta} = 0$.



We check if the new set of filters keep the same frequency peaks by looking at the derivative of $\widetilde{\psi}_l$ against $\beta$ around the frequency peaks. With the notation $\partial_\beta \cdot = \frac{\partial \cdot}{\partial \beta}$ we get:

$$\partial_\beta \widetilde{\psi}_l(\nu) = \frac{\partial_\beta \psi_l}{G} \left( 1 - \left| \frac{\psi_l}{G} \right|^2 \right) - \sum_{j \neq l} \left| \frac{\psi_j}{G} \right|^2 \frac{\partial_\beta \psi_j}{G} \tag{A7}$$

$$= \frac{\psi_{a_l}(\nu)}{G} \log(\nu/\nu_l) - \sum_j \left| \frac{\psi_{a_j}}{G} \right|^3 \log(\nu/\nu_j). \tag{A8}$$

At $\nu = \nu_l = \frac{1}{a_l}(\beta/\gamma)^{1/\gamma} = \frac{2^{-l/Q}}{a_0}(\beta/\gamma)^{1/\gamma}$, with $a_l = a_0 2^{l/Q}$,

$$\partial_\beta \widetilde{\psi_{a_l}}(\nu_l) = -\sum_{j \neq i} \left| \frac{\psi_{a_j}}{G}(\nu_l) \right|^3 \log(\nu_l/\nu_j) \tag{A9}$$

$$= -\frac{\log 2}{Q} \sum_{j \neq l} \left| \frac{\psi_{a_j}}{G}(\nu_l) \right|^3 (l - j). \tag{A10}$$

Around the $l^{\text{th}}$ wavelet, the effect of wavelet neighbors on the frequency localisation of the frequency peak compensate each other, and increasing the resolution $Q$ decreases directly (through factor $\frac{1}{Q}$) and indirectly (by increasing $G$ in $1/G^3$) the impact of the normalisation. We expect however important modifications of the frequency peaks at the borders at high and low frequency where less wavelets are present.

**Impact on wavelet time-deviation**: the time-deviation of a wavelet characterizes how much it is concentrated in the time domain, thus it is an effective measure of its scale. The time deviation of a wavelet $\psi_\xi(t)$ (here in the time domain) can be computed using:

$$\sigma_l = \frac{1}{T} \sqrt{\frac{\int_{-T/2}^{T/2} |\psi_l(t)|^2 t^2 dt}{\int_{-T/2}^{T//2} |\psi_l(t)|^2 dt}}, \tag{A11}$$

which is normalised by the maximum time support of the wavelet $T$.

On Fig. A1 we show the impact of the normalisation on the frequency peaks and on the time-deviations while increasing the resolution Q. Except at the borders, i.e. at low or high frequencies, the normalisation has limited effect on the frequency peaks location and on the time-deviations if the resolution is high enough. It is then acceptable to use the theoretical frequency peaks of the original wavelets without normalisation to establish a proxy for measuring their scale (up to an unknown constant).



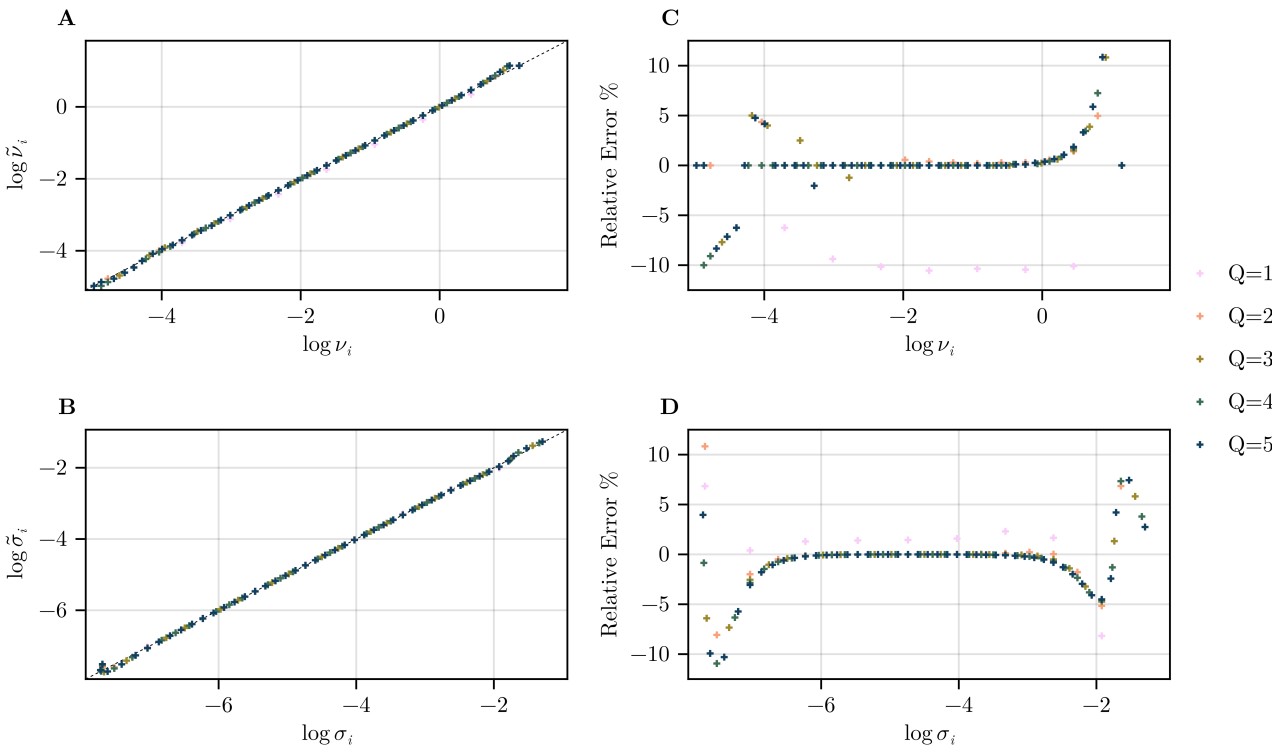

**Figure A1.** Effect of the self-dual normalisation (Eq. 17) on wavelet frequency peaks and time deviations. Top-Left: frequency peaks $\{\nu_i\}_i$ in log scale before ($\nu_i$, x-axis) and after ($\tilde{\nu}_i$, y-axis) normalisation. Top-Right: relative error of the frequency peaks after normalisation. Bottom: the same analysis but with time-deviations

## A3 Statistics of sensible and latent heat fluxes



(a) Sensible heat fluxes

(b) Latent heat fluxes

**Figure A2.** Same as Fig. 5 but for sensible and latent heat fluxes.



## A4 Effect of the turbulence mask on fluxes distributions

In Fig. A3 and Fig. A4, we show the effect of the turbulence mask on the probabilities densities of fluxes in day and night
560   conditions, respectively. We look in particular at the difference between the densities with and without the turbulence mask. We
remark the following: (1) the application of the turbulence mask not only removes regions with low $\tau_w$, i.e. below the threshold
$\delta_\tau = 10^{-3}\,\mathrm{m^2\,s^{-2}}$, but the estimated time-varying spectral gap used to form the turbulence mask also helps in excluding high
amplitude $\tau_w$ regions at low frequencies around $\eta = 10^{-4}$. This is particularly visible in Fig. A4 at nighttime conditions. This
demonstrates that the proposed method reduces the influence of large scale processes for estimating eddy fluxes which are
565   located approximately in the frequency bands $\eta = 10^{-2}$ to $\eta = 10^2$ during day and in the bands $\eta = 1$ to $\eta = 10^2$ during night.
These frequency bands are here mostly preserved by the application of the mask except for the high end of the spectrum
around $\eta = 10^2$ in daytime condition where we transition from the inertial subrange occupied by the turbulent eddies into the
dissipation range; (2) In day and night conditions, a weak and centered sensible heat flux is rejected while a strong positive
(day) or negative (night) sensible heat flux is kept which suggests that the turbulent exchange of heat by eddies is preserved. (3)
570   During the day, weak and centered latent heat and carbon fluxes are removed through filtering while strong latent heat fluxes
and strong carbon uptake, likely linked to evapotranspiration and photosynthesis, resp., are preserved. During the night, strong
negative latent heat and carbon fluxes are excluded by the turbulence mask while positive latent heat and carbon fluxes are
preserved. This suggests that the turbulence identification correctly preserves the processes at play (respiration, photosynthesis
and evapotranspiration) in the turbulent exchange of water vapor and carbon between the forest and the atmosphere.





**Figure A3.** Effect of the application of the turbulence mask on the distributions of time and frequency decomposed fluxes during 8 h daytime periods from 10 to 19 June 2022. Probability densities of the fluxes without turbulence mask filtering ($\mathbb{P}$, first column), with turbulence mask filtering ($\mathbb{P}_{\mathcal{X}}$, second column) and their differences ($\mathbb{P}_{\mathcal{X}} - \mathbb{P}$, third column) where negative values show where data has been removed by turbulence mask filtering.





**Figure A4.** Same as Fig. A3 but during 8h periods at night from 10 to 19 June 2022





## A5 Approximations in the decomposition of the advective term

We analyse here the viability of the approximations made in Eq. 6 for decomposing the advective term. We look at the relative squared error (RSE)

$$\mathrm{RSE}(t) = \frac{(\widetilde{[ws]_\phi} - [ws]_\phi)^2(t)}{\sigma^2_{[ws]_\phi}}, \tag{A12}$$

where $\widetilde{[ws]_\phi}$ is an estimator of $[ws]_\phi$, and $\sigma_{[ws]_\phi}$ is the standard deviation of the averaged advective term over $24\,\mathrm{h}$. We look at the viability of "multiple bands decomposition" estimator composed of wavelets filters:

$$\widetilde{[ws]_\phi} = \sum_{l=1}^{L} [w_l s_l]_\phi, \tag{A13}$$

with $L$ filters as explained in Sec. 2.4

Here, $\phi$ is a Gaussian window whose averaging length is controlled by its variance and the target value is the averaged advective term $[ws]_\phi$.

In Fig. A5, we show the influence of the averaging length, the number of inter-octave bands $Q$ and one of the wavelet shape parameter $\beta$ in the approximation error. We study the advective term for vertical kinematic and sensible heat fluxes. We observe that the error is high with small averaging length and rapidly decreases with increasing averaging length, reaching $10^{-3}\,\sigma_{[ws]_\phi}$ for vertical kinematic flux and $10^{-2}\,\sigma_{[ws]_\phi}$ for sensible heat with 2 hours averaging. The averaged advective term converges to the global flux with large averaging length, which is quantity conserved by the multiple bands decomposition. We note that decreasing the number of inter-octaves bands, i.e. low $Q$, decreases slightly the error because the number of inter-correlation between frequency bands being ignored is reduced. The shape parameter $\beta$ also has an influence on the approximation. We note that decreasing its value reduces the error. This may be due to the shape of the overall turbulence co-spectrum which might be better captured with reasonably low $\beta$ values.





**Figure A5.** Relative squared error in the approximation of the averaged avective term for the multiple band decompositions in function of the averaging length (top), the number of inter-octave bands (middle) and the wavelet shape parameter $\beta$ (bottom). The error is averaged across 10 days from 10 to 19 June 2022.



*Author contributions.* GD developed the methodology and the software, curated the data, made the visuals and wrote the manuscript draft.
595 MC, EJ and NB participated in the development of the methodology and the analysis of the data, reviewed and edited the manuscript. MC and EJ acquired the financial support for the project, supervised the research activity, and provided the data from the FR-Hes flux tower.

*Competing interests.* none

*Acknowledgements.* MC and EJ acknowledge support from a grant by the French National Research Agency (ANR, ANR-21-CE02-0033-01). MC, EJ and NB acknowledge support from a grant overseen by ANR as part of the "Investissements d'Avenir" program (ANR-11-600 LABX-0002-01, Lab of Excellence ARBRE). MC acknowledges support from funding by the Scientific Interest Group "Institut des Mathématiques pour la Planète Terre". MC and GD acknowledge support from funding from "La Région Grand Est" within the program "soutien aux jeunes chercheurs".



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
