# Peer review of "Turbulent transport extraction in time and frequency and the estimation of eddy fluxes at high resolution"

_EGUsphere, 2024_

## Referee Comment (RC1)

**General comments**

The paper "Turbulent transport extraction in time and frequency and the estimation of eddy fluxes at high resolution" by Gabriel Destouet et al. (egusphere-2024-3243) proposes a framework for the estimation of eddy fluxes based on cross-scalogram smoothing, which can obtain high-resolution fluxes in time and frequency. This framework has been applied to a beech forest site and is demonstrated the agreement with the standard eddy covariance method.

Overall, the paper presents a novel method for turbulent flux estimation and the context is well-organized with detailed mathematical and physical foundations, which fits the scope of AMT. However, the motivation and the interpretation of the outcomes do not come out clearly. Please see my major concerns below.

1.  My main concern is the advantages of the HR-TM method compared to the standard eddy covariance method, as it increases computational cost and complexity. This paper points out two limitations of the standard eddy-covariance method, that is, limited temporal resolution and limited frequency resolution. The former leads to fluxes unable to characterize fast dynamics, while the latter may introduce non-turbulent contributions and cause potential biases. To overcome these, the study proposes the HR-TM method for the purpose of estimating fluxes with a high time and frequency resolution.
    For the first limitation, you have increased the temporal resolution of fluxes to 10 min (e.g., Figures 2, 4, and 5). However, 10 min is not a high temporal resolution, and the standard eddy covariance method can also do it. You mentioned in Abstract that the HR-TM method can produce high temporal resolution (1 min) fluxes, but why not show it in Results? Besides, there are clear wave (or oscillation) signals in the time series of 10-min fluxes (Figures 2D, 4, and 5). Is this physical or mathematically generated? I speculate that this oscillation is a mathematical bias introduced by your method. Therefore, the improvement in temporal resolution remains to be clarified.

For the second limitation, the HR-TM method can decompose fluxes into multiple frequency bands and remove the contribution of non-turbulence using the turbulence mask (e.g., Figure 2C) to improve the accuracy of turbulent fluxes. However, the fluxes estimated using the HR-TM method generally agree with that obtained by the standard, especially the comparison in Figures 4 and A2. Therefore, the advantage of the HR-TM method in identifying turbulent transport may be only demonstrated under specific conditions, such as evaluating the flux of passive scalars (e.g., $CO_2$) and evaluating the vertical transport role of nighttime weak turbulence (Figure A2bC-D). Perhaps you can focus on these conditions to demonstrate the practicality of the HR-TM method.

In my understanding, the main advantage of the HR-TM method is that it can obtain high-resolution fluxes in time and frequency, which can be used to identify turbulent coherent structures (such as microfronts and thermal plumes) and to characterize the role of different coherent structures in the vertical turbulent transport. However, it was not highlighted in the application cases of the HR-TM method.

2. My second concern is the average time and temporal resolution. There are 30 min, 10 min, and 1 min mentioned in the paper, but they are not clearly stated. This would make readers confused. Please see my specific comments 4 &5. It is suggested to indicate the corresponding temporal resolution in each step of Figure 1, instead of the general time variable $t$ which is interpreted as $t = kT_c$. Additionally, is it a great improvement that the average time of turbulence mask is different from that of flux estimation? If not, it is suggested to choose the same average time or to state them more clearly.

**Specific comments**

1. L3, "fluxes unable to characterise fast dynamics (< 30min) …" I think the expression is not appropriate. Although the average time of 30 min is commonly used to calculate turbulent fluxes by the eddy covariance method, it is not the only one. For weak turbulence, a smaller average time can be adopted, such as 10 min

and 5 min. This practice can also reduce the amount of data discarded because of failing quality tests. If your method can obtain fluxes with a temporal resolution of ~ 1 minute, it possibly can characterize fast dynamics. So, it may be better to change "<30min" to "~1 min", if possible.

2. L257, "… so that most of the sensible heat $F_H$ is preserved" –Why? or what percentage does "most" correspond to in your quantification?

3. L284, "…, and time averaged to form fluxes resolved in time and frequency" – What is the temporal resolution of flux chosen in this study? 10 min?

4. L292-293, There are great doubts about the selection of the average time. Why can't the average parameters of turbulence mask and flux estimation be the same? If I understand correctly, the fluxes, $F_u$, $F_v$, and $F_w$, are first calculated with a temporal resolution of 30 min (Step B) to obtain the turbulence mask (Step C), then they have to be calculated again with a temporal resolution of 10 min if we want to gain an estimation of kinematic fluxes (Step D)? or $\sigma = 10$ min only is intended to estimate scalar fluxes? Even so, if a grid $(t^*,\ \eta^*)$ is classified as non-turbulent, there will be three grids of flux $F_c(t,\eta)$ being discarded, i.e., $(t^* - T_c,\ \eta^*)$, $(t^*,\ \eta^*)$, and $(t^* + T_c,\ \eta^*)$. Why not use $\sigma = 10$ min directly for the turbulence mask? And you said "…, after step C in Fig. 1, fluxes in time and frequency coordinates have a temporal resolution of 20 Hz" in L318, but the variable $t$ in $F(t,\eta)$ is interpreted as $t = kT_c$ where $T_c$ is the averaging time (L102). Please clarify the choice of average time.

5. Figure 2, again about the selection of the average time. Is it right that the temporal resolution of Figures 2A and 2B is 30 min while the temporal resolution of Figures 2C and 2D is 10 minutes? This specially-designed different average time makes readers confused.

6. L357, there is no dotted line in Figure 2B. It might be also helpful to plot the dotted line in Figure 2A.

7. L374-375, "If the turbulence mask does not cover any frequency bands at a given time, i.e. no turbulence is detected at that time, the calculated flux is undefined". If there is no turbulence mask, can we consider the turbulent flux to be zero?

8. L379-380, " The evolution of $\tau_w$ against $\eta$ presents similar characteristics as the spectrum of the vertical velocity and as the cospectrum of $u \cdot w$". In my understanding, the characteristics of the $w$ spectrum and $uw$ cospectrum are different (e.g., the slope in the inertial subrange), so how does the evolution of $\tau_w$ show similar characteristics as the two?

9. Figure 3, what is the physical meaning of red crosses in Figure 3? It is not mentioned in the main text.

10. Figure 4, which method do you adopt to calculate u*? standard eddy covariance method or cross-scalogram smoothing method?

**Technical corrections**

1. L101, "$N$ is the number of averaging periods ", but there is no variable $N$ in Eq. 2.

2. L230, "Reynold's frozen" – Is that might to be Taylor's frozen?

3. L260, "…, that means in removes noise"? – There may be typing errors, please check.

---

## Author Comment (AC1)

**Turbulent transport extraction in time and frequency and the estimation of eddy fluxes at high resolution**

Gabriel Destouet[*1], Nikola Besic[2], Emilie Joetzjer[1], and Matthias Cuntz[1]

[1]UMR SILVA, INRAE, AgroParisTech, Université de Lorraine, 54280 Champenoux, France

[2]IGN, ENSG, Laboratoire d'inventaire forestier (LIF), 54000 Nancy, France

[*]*Corresponding author: gabriel.destouet@inrae.fr*

Dear reviewers, thank you very much for your very stimulating questions, comments and suggestions. We did our best to address them appropriately in the revised version of the manuscript, and by doing so to improve the quality of the presented research.

The remarks and comments are enumerated and denoted in font **R#**.**C#**. We use blue to refer to sections and figures in the revised manuscript.

For each reviewer, we provide the same general response addressing a global concern that both reviewers appear to have raised. This is followed by a specific response to the reviewer, and then we address each of their comments individually.

**Reviewer 1 (R1)**

**General comments:**

The paper "Turbulent transport extraction in time and frequency and the estimation of eddy fluxes at high resolution" by Gabriel Destouet et al. (egusphere-2024-3243) proposes a framework for the estimation of eddy fluxes based on cross-scalogram smoothing, which can obtain high-resolution fluxes in time and frequency. This framework has been applied to a beech forest site and is demonstrated the agreement with the standard eddy covariance method.

Overall, the paper presents a novel method for turbulent flux estimation and the context is well-organized with detailed mathematical and physical foundations, which fits the scope of AMT. However, the motivation and the interpretation of the outcomes do not come out clearly. Please see my major concerns below.

⟹ We would like to thank the reviewer for its insightful comments, which have helped us clarify the manuscript. Below, you will find our general response addressing common remarks made by the reviewers, as well as a specific response to the reviewer's detailed comments on our work.

**General response:** One of the primary issues in our initial version was the lack of emphasis on the distinction between "technical" and "physical" time resolutions. The former refers to the time step of the estimations, while the latter relates to the averaging time. We acknowledge that we have not been differentiating clearly enough these concepts in the original paper, which may have obscured the understanding of the proposed approach.

Our motivation for the method arose from studying eddy-fluxes with a faster rate of change. Specifically, we aimed to increase the "physical" time resolution of the flux estimations by reducing the averaging time. However, the standard eddy-covariance method does not allow for adjusting the averaging time without affecting the filtering of perturbative scales. To address this, we propose decoupling the filtering of perturbative scales from the flux calculations. This led us to develop a method based on time-frequency analysis, particularly wavelet analysis, that enables:

1. Identifying turbulent coherent structures by tracking turbulent spectra across the time-frequency domain.

2. Freely adjusting the averaging time, allowing for estimation of eddy fluxes with high time resolution.

The first point involves creating a time-frequency map of the turbulent structures by analyzing the Reynold's tensor and establishing the turbulent mask. The second point follows from the first; with accurate isolation of turbulent structures, one can adjust the averaging time to correctly resolve their contribution to turbulent transport.

Additionally, since our method allows for high "technical" resolution, we suggested it could enhance data availability after applying standard quality tests. However, as both reviewers noted, the standard eddy-covariance method can also achieve high "technical" resolution, although this is less common in standard implementations. We have removed any references to increasing the "technical" resolution from the original paper and focused solely on our original motivation: increasing the "physical" resolution without compromising the quality of flux estimations.

**Specific Response:** As remarked by the reviewer, this method present a higher complexity and computational cost. However, we believe these costs should be considered in light of the advantages of the proposed approach,

particularly its ability to more precisely identify turbulent structures and remain robust and adaptive to changing conditions when estimating fluxes. To be practically useful for standard flux calculations, the proposed method would benefit from a reduction in computational cost. We outline some possible improvements in R1.C1 and in the discussion section of the revised manuscript. In its current state, we believe that the proposed method and its implementation would be a valuable addition to the community, opening up new research perspectives, such as higher-frequency flux estimates or improved turbulent structure identification, as suggested by the reviewer.
* * *
**R1.C1: My main concern is the advantages of the HR-TM method compared to the standard eddy covariance method, as it increases computational cost and complexity. This paper points out two limitations of the standard eddy-covariance method, that is, limited temporal resolution and limited frequency resolution. The former leads to fluxes unable to characterize fast dynamics, while the latter may introduce non-turbulent contributions and cause potential biases. To overcome these, the study proposes the HR-TM method for the purpose of estimating fluxes with a high time and frequency resolution. For the first limitation, you have increased the temporal resolution of fluxes to 10 min (e.g., Figures 2, 4, and 5). However, 10 min is not a high temporal resolution, and the standard eddy covariance method can also do it. You mentioned in Abstract that the HR-TM method can produce high temporal resolution (1 min) fluxes, but why not show it in Results? Besides, there are clear wave (or oscillation) signals in the time series of 10-min fluxes (Figures 2D, 4, and 5). Is this physical or mathematically generated? I speculate that this oscillation is a mathematical bias introduced by your method. Therefore, the improvement in temporal resolution remains to be clarified.**

$\implies$ In the revised manuscript, particularly in Sec. 2.6, we clarify the distinction between the estimation time-step (sub-sampling) and the averaging time, i.e., between a "technical" and "physical" resolution. Additionally, as suggested by the reviewer, we present additional fluxes with a 1-minute time sampling, resulting from averaging with 1, 10, and 30-minute windows. These are shown in Fig. 2.

We acknowledge the reviewer's observation that the proposed approach increases the computational cost of the estimation. With the current implementation (v0.1.0 of TurbulenceFlux.jl), on an Intel i9-12950HX CPU with 32GB of RAM, the calculation of a 24-hour cross-scalogram at 20Hz requires $73\pm2$ seconds and approximately 16GB of RAM. The full estimation of the fluxes (kinematic, $CO_2$, sensible and latent heat), as presented in Fig. 1, takes around 10 minutes for 24 hours. These computational costs are detailed in the discussion section of the manuscript.

We believe this cost is comparable to current estimation software. The computational cost could be significantly reduced by:

- Limiting the number of scales analyzed, assuming, for example, that everything above $\eta=1$ is part of the turbulent transport.

- Parallelizing computations on the GPU, especially since cross-scalograms are computed using convolutions.

Finally, we argue that oscillations in the estimated signals are inherent to filtering procedures. These oscillations are also observed in signals estimated using the eddy-covariance method (e.g., Fig. 2). Both methods rely on filters that assume the turbulent process is confined within a specific frequency band, which may not fully capture the

turbulent process, leading to oscillations in the estimations. The intermittent nature of turbulent transport and the influence of the averaging time could also explain these oscillations. In Fig. 2, we show three different eddy-fluxes with different averaging times. The fluxes estimated with our method with the 30-minute averaging time present fewer oscillations than the 30-minute eddy-covariance method. This difference can be attributed to the use of a Gaussian window in our method, whereas the standard eddy-covariance method employs a rectangular averaging window, which is less effective in filtering high frequencies.

**R1.C2: For the second limitation, the HR-TM method can decompose fluxes into multiple frequency bands and remove the contribution of non-turbulence using the turbulence mask (e.g., Figure 2C) to improve the accuracy of turbulent fluxes. However, the fluxes estimated using the HR-TM method generally agree with that obtained by the standard, especially the comparison in Figures 4 and A2. Therefore, the advantage of the HR-TM method in identifying turbulent transport may be only demonstrated under specific conditions, such as evaluating the flux of passive scalars (e.g., CO2) and evaluating the vertical transport role of nighttime weak turbulence (Figure A2bC-D). Perhaps you can focus on these conditions to demonstrate the practicality of the HR-TM method. In my understanding, the main advantage of the HR-TM method is that it can obtain high-resolution fluxes in time and frequency, which can be used to identify turbulent coherent structures (such as microfronts and thermal plumes) and to characterize the role of different coherent structures in the vertical turbulent transport. However, it was not highlighted in the application cases of the HR-TM method.**

$\Longrightarrow$ The principal objective of the results section was to demonstrate that the HR-TM method, with its novel approach to identifying turbulence across the time-frequency domain, is:

1. Consistent with the standard eddy-covariance method during the day. We aimed to show that at the Hesse site, the HR-TM method could produce fluxes comparable to those from the standard eddy-covariance method with a 30-minute averaging time. In unstable stratification, this averaging time is well-suited for this site, and establishing this consistency was crucial for validating the reliability of the HR-TM method.

2. Adaptive for estimating fluxes under various conditions. The filtering of perturbative scales in our method is conditioned on the presence or absence of turbulent processes, as illustrated by the turbulence mask. In contrast, the eddy-covariance method relies on a pre-defined averaging time to localize the spectral gap, which must be adjusted for varying conditions. In Fig. 2C, the solid line is the standard cut-off which is constant across the day (note that its slight variation is due to the coordinate system, where the normalized frequency changes with the wind amplitude). In comparison, our method adapt its filtering both in unstable and stable localization with the turbulent mask, the dashed line in Fig.2C shows the adaptative spectral gap which is applied on top of the turbulent mask to remove potential trends in low frequencies, this can be seen as the adaptive variant of the standard cut-off.

While we agree with the reviewer that turbulence identification may hint at the analysis of coherent structures, we believe that significant technical and experimental work would be required to effectively use the turbulence mask for characterizing their role. In particular, if one want to retrieve the velocity and scalar time-series associated with a particular structure. This is due to the fact that our method for identifying the turbulent structure operates on the

product of the time-series and not on the time-series themself, hence it is not straightforward to apply the turbulent mask on the individual time-series to identify some patterns associated to these structures. We think this should be addressed in a separate work.

**R1.C3: My second concern is the average time and temporal resolution. There are 30 min, 10 min, and 1 min mentioned in the paper, but they are not clearly stated. This would make readers confused. Please see my specific comments 4 &5. It is suggested to indicate the corresponding temporal resolution in each step of Figure 1, instead of the general time variable t which is interpreted as t = kTc. Additionally, is it a great improvement that the average time of turbulence mask is different from that of flux estimation? If not, it is suggested to choose the same average time or to state them more clearly.**

$\Longrightarrow$ The time step for all fluxes computed with our method (in cross-scalogram form or time signals) is 1 minute. Our motivation for using different averaging times for the turbulence mask and flux estimation was primarily driven by:

1. Demonstrating that these two steps are performed independently. Establishing the turbulent mask identifies the relevant time-frequency regions occupied by turbulent coherent structures. Resolving fluxes in the time-frequency domain by computing their smooth cross-scalograms quantifies the fluxes at each time-frequency point. Combining both steps achieves the flux estimation.

2. With the current method for establishing the turbulent mask, we found it beneficial to slightly overestimate the time-frequency region containing the turbulent coherent structures. This ensures that during flux estimation, we do not miss any quantities. Therefore, we chose a higher averaging time for the turbulence mask than for resolving fluxes in the time-frequency space.

For the distinction between the estimation time-step and the averaging time see our general response and Sec. 2.6 of the revised manuscript. The Sec. 2.6 also explain why a different averaging time, in particular a larger one, is used for establishing the turbulence mask.

**Specific comments**
* * *
**R1.C4: L3, "fluxes unable to characterise fast dynamics (< 30min) . . . " I think the expression is not appropriate. Although the average time of 30 min is commonly used to calculate turbulent fluxes by the eddy covariance method, it is not the only one. For weak turbulence, a smaller average time can be adopted, such as 10 min and 5 min. This practice can also reduce the amount of data discarded because of failing quality tests. If your method can obtain fluxes with a temporal resolution of ∼ 1 minute, it possibly can characterize fast dynamics. So, it may be better to change "<30min" to " 1 min", if possible.**

$\Longrightarrow$ We agree with the reviewer that under weak turbulence conditions, the averaging time for standard eddy-covariance can be reduced, allowing for higher "physical" resolution of fluxes. In strong turbulence, however, the averaging time is constrained by the spectral gap's localization, and reducing it further would affect the filtering of perturbative scales. For clarification on the distinction between "technical" and "physical" time resolution, please see general response and Sec. 2.6. We also changed the abstract extensively and the statement should be much clearer

now.

**R1.C5: L257, ". . . so that most of the sensible heat FH is preserved" –Why? or what percentage does "most" correspond to in your quantification?**

$\implies$ The threshold $\delta_\tau$ was set manually by examining how the distribution of sensible heat deviates as the amplitude of $\tau_w$ increases. This relationship is illustrated in Fig. 3, where an increase in the amplitude of $\tau_w$ corresponds to a higher empirical probability of heat exchange occurring (either positive or negative, depending on the stratification). The departure was found around $\tau_w = 1e^{-3}$.

We modified the manuscript accordingly in Sec. 2.5.

**R1.C6: L284, ". . . , and time averaged to form fluxes resolved in time and frequency" – What is the temporal resolution of flux chosen in this study? 10 min?**

$\implies$ The time-step is always 1 min and the averaging time is varied to conditions the "physical" resolution of the flux estimates. See our general response and the details in Sec. 2.6.

**R1.C7: L292-293, There are great doubts about the selection of the average time. Why can't the average parameters of turbulence mask and flux estimation be the same? If I understand correctly, the fluxes, Fu, Fv, and Fw, are first calculated with a temporal resolution of 30 min (Step B) to obtain the turbulence mask (Step C), then they have to be calculated again with a temporal resolution of 10 min if we want to gain an estimation of kinematic fluxes (Step D)? or $\sigma = 10$ min only is intended to estimate scalar fluxes? Even so, if a grid $(t^*, \eta^*)$ is classified as nonturbulent, there will be three grids of flux $F_c(t, \eta)$ being discarded, i.e., $(t^* - T_c, \eta^*), (t^*, \eta^*)$ and $(t^* + T_c, \eta^*)$. Why not use $\sigma = 10$ min directly for the turbulence mask? And you said ". . . , after step C in Fig. 1, fluxes in time and frequency coordinates have a temporal resolution of 20 Hz" in L318, but the variable t in $F(t, \eta)$ is interpreted as t = kTc where Tc is the averaging time (L102). Please clarify the choice of average time.**

$\implies$ Please see our answer to R1.C3 and our general response. The information concerning the time-grid and the different averaging times is now present in Sec. 2.6.

**R1.C8: Figure 2, again about the selection of the average time. Is it right that the temporal resolution of Figures 2A and 2B is 30 min while the temporal resolution of Figures 2C and 2D is 10 minutes? This specially-designed different average time makes readers confused.**

$\implies$ In all figures, the time step is 1 minute. Figure 2A is obtained with a 30-minute averaging time. Figure 2B is the second derivative of 2A. Figure 2C is obtained with a 10-minute averaging time.

Please see our answer to R1.C3 , and modifications made in Sec. 2.6 and in the text describing Fig. 2 in Sec. 3.1.

**R1.C9: L357, there is no dotted line in Figure 2B. It might be also helpful to plot the dotted line in Figure 2A.**

$\implies$ Thanks for the suggestion. The dotted line is now shown in Fig. 2.

**R1.C10: L374-375, "If the turbulence mask does not cover any frequency bands at a given time, i.e. no turbulence is detected at that time, the calculated flux is undefined". If there is no turbulence mask, can we consider the turbulent flux to be zero ?**

$\implies$ In practice, we observed over the turbulent scales that regions not covered by the turbulent mask contain vanishing fluxes. This is expected when no coherent structures are present to vertically mix the air layers. In this

work, we prefer to consider that the flux is undefined since we integrate the flux over an empty frequency domain. However as remarked by the reviewer, in this context of eddy flux estimation, we could indeed consider that the turbulent flux is zero. We updated the manuscript accordingly and added that "When estimating ecosystem fluxes, this is a typical scenario where it is important to account for the storage term."

**R1.C11: L379-380, " The evolution of $\tau w$ against $\eta$ presents similar characteristics as the spectrum of the vertical velocity and as the cospectrum of $uw$". In my understanding, the characteristics of the w spectrum and $uw$ cospectrum are different (e.g., the slope in the inertial subrange), so how does the evolution of $\tau w$ show similar characteristics as the two?**

$\Longrightarrow$ We thank the reviewer for noticing this imprecision. The main characteristic shared by $\tau w$, the $w$ spectrum, and the $uw$ spectrum is the change in the frequency bandwidth occupied by the turbulent scales between unstable and stable stratification. This characteristic is seen in Figure 3.

We revised the manuscript accordingly in Sec. 3.2.

**R1.C12: Figure 3, what is the physical meaning of red crosses in Figure 3? It is not mentioned in the main text.**

$\Longrightarrow$ The red crosses show the empirical probability distribution of heat exchange occuring (either positive or negative). Please see our response to R1.C5 and the corrections made in Sec. 3.2.

**R1.C13: Figure 4, which method do you adopt to calculate u*? standard eddy covariance method or cross-scalogram smoothing method?**

$\Longrightarrow$ The $u^*$ metric is calculated using $\left( \overline{u'w'}^2 + \overline{v'w'}^2 \right)^{1/4}$ with the standard eddy-covariance for estimating the kinematic fluxes.

This is now indicated in the revised manuscript in Sec. 3.3.

**Technical corrections**
* * *
**R1.C14: L101, "N is the number of averaging periods ", but there is no variable N in Eq. 2.**

$\Longrightarrow$ Corrected

**R1.C15: L230, "Reynold's frozen" – Is that might to be Taylor's frozen?**

$\Longrightarrow$ Corrected

**R1.C16: L260, "..., that means in removes noise"? – There may be typing errors, please check.**

$\Longrightarrow$ Corrected

---

## Author Comment (AC2)

**Turbulent transport extraction in time and frequency and the estimation of eddy fluxes at high resolution**

Gabriel Destouet[*1], Nikola Besic[2], Emilie Joetzjer[1], and Matthias Cuntz[1]

[1]UMR SILVA, INRAE, AgroParisTech, Université de Lorraine, 54280 Champenoux, France

[2]IGN, ENSG, Laboratoire d'inventaire forestier (LIF), 54000 Nancy, France

[*]*Corresponding author: gabriel.destouet@inrae.fr*

Dear reviewers, thank you very much for your very stimulating questions, comments and suggestions. We did our best to address them appropriately in the revised version of the manuscript, and by doing so to improve the quality of the presented research.

The remarks and comments are enumerated and denoted in font **R#**.**C#**. We use blue to refer to sections and figures in the revised manuscript.

For each reviewer, we provide the same general response addressing a global concern that both reviewers appear to have raised. This is followed by a specific response to the reviewer, and then we address each of their comments individually.

**Reviewer 2 (R2)**

**General comments**

**The manuscript "Turbulent transport extraction in time and frequency and the estimation of eddy fluxes at high resolution" by Gabriel Destouet et al. (egusphere-2024-3243) presents a method for determining if the turbulence is sufficiently developed considering time and the eddy scale.**

**The manuscript is clear. I appreciated the progressiveness of the introduction that guides well the reader and found that the technical parts in the methods are well explained. It is in Atmospheric Measurement Techniques' scope and it brings clear contributions to the flux community. A few points could be clearer, please see my major concerns below [...] The scientific approach and methods applied are valid and the results and discussion well based on it. Overall, I believe that this manuscript represents a significant contribution to the field of atmospheric measurement techniques and I recommend it for publication in Atmospheric Measurement Techniques.**
* * *
We would like to thank the reviewer for their review and positive feedback on our manuscript. Below, you will find our general response addressing common remarks made by the reviewers, as well as a specific response to the reviewer's detailed comments on our work.

**General response:** One of the primary issues in our initial version was the lack of emphasis on the distinction between "technical" and "physical" time resolutions. The former refers to the time step of the estimations, while the latter relates to the averaging time. We acknowledge that we have not been differentiating clearly enough these concepts in the original paper, which may have obscured the understanding of the proposed approach.

Our motivation for the method arose from studying eddy-fluxes with a faster rate of change. Specifically, we aimed to increase the "physical" time resolution of the flux estimations by reducing the averaging time. However, the standard eddy-covariance method does not allow for adjusting the averaging time without affecting the filtering of perturbative scales. To address this, we propose decoupling the filtering of perturbative scales from the flux calculations. This led us to develop a method based on time-frequency analysis, particularly wavelet analysis, that enables:

1. Identifying turbulent coherent structures by tracking turbulent spectra across the time-frequency domain.

2. Freely adjusting the averaging time, allowing for estimation of eddy fluxes with high time resolution.

The first point involves creating a time-frequency map of the turbulent structures by analyzing the Reynold's tensor and establishing the turbulent mask. The second point follows from the first; with accurate isolation of turbulent structures, one can adjust the averaging time to correctly resolve their contribution to turbulent transport.

Additionally, since our method allows for high "technical" resolution, we suggested it could enhance data availability after applying standard quality tests. However, as both reviewers noted, the standard eddy-covariance method can also achieve high "technical" resolution, although this is less common in standard implementations. We have removed any references to increasing the "technical" resolution from the original paper and focused solely on our original motivation: increasing the "physical" resolution without compromising the quality of flux estimations.

**Specific Response:** As the reviewer mentioned, our method is based on previous research on the wavelet analysis of turbulence. In this paper, we aimed to address some of the common mathematical and technical challenges that come up with wavelet transforms, especially when it comes to energy conservation and practical use. We think that the proposed framework, which uses generalized Morse wavelets, will be a great addition to the community. It offers both the flexibility and the physical guarantees needed for future research to dive deeper into analyzing and isolating turbulent structures.

**R2.C1: The manuscript seems to suggest an innovation is calculating fluxes at smaller time periods. Estimating fluxes at high resolution can be done using standard eddy-covariance. The time and frequency average are commonly the same but there is no reason why not to calculate the instantaneous deviation (c') using the 30-min average and then averaging w'c' every 1 min. Conserving thus lower than 1-min frequency while having the 1 min fluxes calculated. Also calculating 1 min flux using continuous wavelets has been done in Schaller et al. (2017).**

$\Longrightarrow$ We agree with the reviewer. We have removed the claim from the manuscript that the method produces fluxes with shorter time steps. The abstract, discussion, and conclusion have been revised to focus on the two main features we consider important: (1) the identification of turbulent transport and (2) the ability to change the averaging time without impacting the filtering.

**R2.C2: The fact that Schaller et al. (2017) have used wavelets for addressing short turbulent events could be cited in the introduction when enumerating the uses of wavelets. The authors justify choosing the Generalised Morse Wavelets to avoid making an arbitrary choice of a particular family of wavelets such as Mexican hat or Morlet wavelets. Then it presents the parameters used. It seems the Generalised Morse Wavelet is of a "more flexible use", but once you define its parameters, it is hard to understand how different this is from choosing another particular wavelet.**

$\Longrightarrow$ The revised manuscript now also references the work of Schaller et al. (2017) for identifying short turbulent events.

One of the primary challenges with wavelet transforms is the variability that can arise from switching between different wavelet families and the associated implementation costs. The advantage of Generalized Morse Wavelets lies in their flexibility, Their spectrum can be easily adjusted to various forms by tweaking their parameters. This flexibility also allows us to see how estimations are affected by wavelet parameters, which could lead to new research paths for studying turbulent transport and improving the estimation process.

The proposed framework is not restricted to using this specific wavelet family. It is sufficiently general to use any set of filters that can decompose frequencies and separate different scales. In this study, we employ Generalized Morse Wavelets as we find that they currently offer the most practical approach for implementing the proposed framework.

**R2.C3: The manuscript would benefit from showing clearly an equation stating how to calculate the global flux from wavelets. The result from using eq. 15-17 together I imagine.**

$\Longrightarrow$ The manuscript now includes a brief paragraph in Sec. 2.6 summarizing the steps to calculate the flux at a time t from the time-frequency decomposition of the flux.

**R2.C4: The terms "smoothing" and "local smoothing" are used several times in the manuscript and**

**seems to be one of the key concepts. Once it includes a citation "The local smoothing of cross-scalograms (Mauder et al., 2007)", however the authors in Mauder et al. (2007) do not employ the term themselves. In the manuscript it seems to be simply a time average, so how does it differ from it and is it really relevant to employ this wording? The risk is to lose the reader with too technical terms.**

$\Longrightarrow$ The term originates from the work of Torrence and Compo (1998) on the analysis of turbulence using wavelets. Mauder et al. (2007) built upon this work, which led us to retain the term "smoothing" as defined in Section 5 of Torrence and Compo (1998). The term "smoothing" is more general, encompassing various averaging windows beyond the traditional rectangular one. In Fig. 2 of the revised manuscript, a difference can be seen, where EC30 is using a rectangular window while HRTM uses a Gaussian window with $\sigma = 30$ min. The estimation with the Gaussian window has fewer high-frequency components.

**R2.C5: The figures showing the time and frequency decomposed fluxes with a mask on top are well-appreciated. They are clear and intuitive. In the conclusion, the authors state that the proposed method "opens up new research perspectives, in particular the analysis of ecosystem response to rapid environmental changes ($<$ 1 hour)." This is an important point, but it could be expanded upon. The sentence seems to suggest the advantage of calculating fluxes for shorter time periods, but this is not the innovative part of the manuscript (see major comment 1). The main contribution lies in doing the analysis in the frequency domain and considering the changing conditions for turbulence flagging.**

$\Longrightarrow$ Thank you for the positive feedback on the figures. Regarding the conclusion, we appreciate your insight and have revised it to better highlight our core contributions. Specifically, we emphasize two key aspects: the decoupling of the filtering process from flux calculations, and the ability to localize turbulent coherent structures across time and frequency. Please see our general response above for further details.

**Minor comments**
* * *
**R2.C6: l.12, 1 min is mentioned but is not shown in the rest of the manuscript.**

$\Longrightarrow$ We now show the diffrent averaging time used in the paper including the 1 min.

**R2.C7: In p.5 l.124-125, the sentence may not be rigorously all correct. In standard eddy covariance, although we commonly only refer to a single averaging time, in reality we do two. One to calculate the instant deviation and another to pass from 10/20Hz to 30-min. See major comment 1.**

$\Longrightarrow$ Please see our general response. In the revised manuscript, regarding the "time resolution" of the estimates, we insist on the ability to vary the averaging time and not on the ability to decrease the time-steps of the estimations.

**R2.C8: In p.5 l.125-127, The sentence does not refer to standard eddy covariance anymore, rephrasing may help avoid misunderstanding. Such as by adding "Alternatively" in the beginning or some other alternative.**

$\Longrightarrow$ Thank you for the proposed reformulation.

**R2.C9: In p.5 l.141, the term "fluxes" repeats 3 times. I suggest to reformulate as such "The advective term is decomposed into turbulent eddy fluxes and other fluxes, encompassing those generated by large-scale processes and noise.".**

$\Longrightarrow$ Thank you for the proposed reformulation.

**R2.C10: In p.8 l.211, is K defined?**

$\Longrightarrow$ The number of frequency bands K is determined during the initialization of the wavelet bank. The definition of K depends on the criterion used to reject poorly sampled wavelets. In our case, we set a low-frequency bound and reject any wavelets whose peaks fall below it.

**R2.C11: In p.8 l.213-214, is it possible that peaks at low frequencies may represent real information and thus be acceptable?**

$\Longrightarrow$ Our assumption is that these are not considered eddy fluxes, as their scales are too large for the study area. Separating the small turbulent scales from the larger ones seems reasonable. The proposed approach still allows for later analysis to assess whether the lower scales contribute to local turbulent transport.

**R2.C12: In p.10 l. 293, "less smoothing" seems imprecise.**

$\Longrightarrow$ Corrected

**R2.C13: In p. 14, l.357, "dotted" should be "dashed".**

$\Longrightarrow$ Corrected